# Differences in BVOC oxidation and SOA formation above and below the forest canopy

Benjamin C. Schulze[1], Henry W. Wallace[1], James H. Flynn[2], Barry L. Lefer[3], Matt H. Erickson[2], B. Tom Jobson[4], Sebastien Dusanter[5,6,7], Stephen M. Griffith[7,‡], Robert F. Hansen[8,^], Philip S. Stevens[8], Timothy VanReken[4,+], Robert J. Griffin[1*]

[1] Department of Civil and Environmental Engineering, Rice University, Houston, TX, 77005

[2] Department of Earth and Atmospheric Sciences, University of Houston, Houston, TX, 77204

[3] Airborne Sciences Program, NASA, Washington, DC, 20546

[4] Department of Civil and Environmental Engineering, Laboratory for Atmospheric Research, Washington State University, Pullman, WA, 99164

[5] Mines Douai, SAGE, F-59508 Douai, France

[6] Université de Lille, 59655 Villeneuve d'Ascq, France

[7] School of Public and Environmental Affairs, Indiana University, Bloomington, IN, USA 47405

[8] Department of Chemistry, Indiana University, Bloomington, IN, USA 47405

[‡]Now at Department of Chemistry, The Hong Kong University of Science and Technology, Kowloon, Hong Kong

[^]Now at School of Chemistry, University of Leeds, Leeds, UK LS2 9JT

[+]Now at National Science Foundation, Washington, DC 22230

[*]Corresponding author: 713-348-2093, rob.griffin@rice.edu

**Abstract**

Gas-phase biogenic volatile organic compounds (BVOCs) are oxidized in the troposphere to produce secondary pollutants such as ozone ($O_3$), organic nitrates ($RONO_2$), and secondary organic aerosol (SOA). Two coupled zero-dimensional models have been used to investigate differences in oxidation and SOA production from isoprene and α-pinene, especially with respect to the nitrate radical ($NO_3$), above and

below a forest canopy in rural Michigan. In both modeled environments (above and below the canopy), $NO_3$ mixing ratios are relatively small (<0.5 pptv); however, daytime (8:00-20:00 LT) mixing ratios below the canopy are two-to-three times larger than those above. As a result of this difference, $NO_3$ contributes 12% of total daytime α-pinene oxidation below the canopy while only contributing 4% above. Increasing background pollutant levels to simulate a more polluted suburban or peri-urban forest environment increases the average contribution of $NO_3$ to daytime below-canopy α-pinene oxidation to 32%. Gas-phase $RONO_2$ produced through $NO_3$ oxidation undergoes net transport upward from the below-canopy environment during the day, and this transport contributes up to 30% of total $NO_3$-derived $RONO_2$ production above the canopy in the morning (~7:00). Modeled SOA mass loadings above and below the canopy ultimately differ by less than 0.5 μg m$^{-3}$, and extremely low-volatility organic compounds dominate SOA composition. Lower temperatures below the canopy cause increased partitioning of semi-volatile gas-phase products to the particle phase and up to 35% larger SOA mass loadings of these products relative to above the canopy in the model. Including transport between above- and below-canopy environments increases above-canopy $NO_3$-derived α-pinene $RONO_2$ SOA mass by as much as 45%, suggesting that below-canopy chemical processes substantially influence above-canopy SOA mass loadings, especially with regard to monoterpene-derived $RONO_2$.

**Keywords:** Nitrate radical, biogenic volatile organic compounds, organic nitrates, secondary organic aerosol

## 1. Introduction

Globally, organic compounds account for a substantial fraction of total atmospheric aerosol mass (Zhang et al., 2007; Jimenez et al., 2009) and therefore have significant implications for health, visibility, and climate. Rather than being emitted directly, nearly 70% of this material is thought to be secondary organic aerosol (SOA) formed from the oxidation of volatile organic compounds (VOCs) (Hallquist et al., 2009). Many of the relevant VOCs are biogenic in origin, causing naturally emitted compounds to contribute substantially to tropospheric aerosol burdens (Seinfeld and Pankow, 2003). Isoprene ($C_5H_8$) and monoterpenes ($C_{10}H_{16}$) are important biogenic VOCs (BVOCs) due to their significant rates of emission and reactivity. Studies suggest that together they comprise 55-65% of non-methane VOC emissions globally (Guenther et al., 1995; Hallquist et al., 2009; Guenther et al., 2012) with estimated yearly emissions of 535 Tg and 157 Tg for isoprene and monoterpenes, respectively (Arneth et al., 2011; Guenther et al., 2012). Thus, characterizing the oxidation chemistry and subsequent formation of SOA from BVOCs, especially isoprene and monoterpenes, is critical. Despite continual progress, many questions remain regarding the mechanisms of SOA production, and current large-scale atmospheric models often under-predict organic aerosol (OA) mass loadings (Heald et. al., 2005; Volkamer et al., 2006; Pye and Seinfeld 2010).

Forests represent a major source of BVOC emissions and therefore heavily influence processes related to ozone ($O_3$) and SOA formation on a global scale. As a result, major efforts have recently been undertaken to better understand the dynamics of chemical processing in forest environments and the related exchange with the atmosphere (Pugh et al., 2010; Steiner et al., 2011; Wolfe et al., 2011; Bryan et al., 2012; Ashworth et al., 2015). These studies have improved understanding of in-canopy gas-phase chemical processing, revealed current problems associated with BVOC degradation schemes, and highlighted the relevance of atmospheric turbulence within the canopy layer; however, aerosol formation processes within forests have thus far received less attention (Ashworth et al., 2015; VanReken et al., 2015). Observed differences in photolysis rates, temperature, and mixing ratios of BVOCs and other trace gases above and below the canopy, combined with measurements that indicate above- and below-canopy environments frequently experience atmospherically decoupled conditions, suggest that chemical processes occurring above and below the forest canopy may be substantially different (Wallace, 2011; Foken et al., 2012), implying the potential for different SOA formation and loss processes.

While differences in $O_3$ and hydroxyl radical (OH) mixing ratios above and below the forest canopy, major daytime oxidants, have been better characterized, the effect of the forest canopy on the nitrate radical ($NO_3$), a known significant nighttime oxidant of BVOCs, has received less study (Fuentes et al., 2007; Bryan et al., 2012; Ashworth et al., 2015). Ambient $NO_3$ concentrations depend strongly on anthropogenic combustion processes, as the compound is formed from the reaction of $O_3$ and nitrogen dioxide ($NO_2$) and lost, either directly or indirectly, through reaction with nitric oxide (NO) and $NO_2$. Daytime $NO_3$ concentrations are generally assumed to be negligible, as rapid photolysis and reaction with NO result in midday lifetimes as low as 5s (Monks, 2005). However, recent work has highlighted the potential for relevant daytime $NO_3$ concentrations (Geyer et al., 2003a; Brown et al., 2005; Fuentes et al., 2007; Pratt et al., 2012). For example, the shade provided by a forest canopy and corresponding reduction in photolysis rates may result in elevated concentrations near the ground relative to the mixed boundary layer above (Brown et al., 2005; Fuentes et al., 2007).

Differences in oxidant mixing ratios above and below the canopy may lead to above/below-canopy differences in the production pathway of organic nitrates ($RONO_2$), which form either through $NO_3$ oxidation of BVOCs or through OH oxidation followed by reaction with NO. Organic nitrates, many of which have volatilities low enough to partition to the aerosol phase, depend on both BVOCs (biogenic) and $NO_x$ (primarily anthropogenic) for their formation, causing these products to often be particularly important to aerosol production in regions subject to high biogenic and anthropogenic emissions (Hallquist et al., 1999; Fry et al., 2009, 2011, 2013, 2014; Rollins et al., 2013; Xu et al., 2014, 2015; Lee et al., 2016). As it has recently become apparent that $RONO_2$ produced through $NO_3$ oxidation of monoterpenes are a particularly important source of SOA (e.g. Fry et al., 2009; Rollins et al., 2013; Fry et al., 2013; Ayres et al., 2015), underestimating formation through this pathway may partially contribute to the current under-prediction of SOA in large-scale models. In addition, by serving as nitrogen oxide ($NO_x = NO + NO_2$) reservoirs, $RONO_2$ species influence $O_3$ production and the oxidative capacity of the atmosphere (Wu et al.,

2007; Farmer et al., 2011; Paulot et al., 2012). As a result, modifications to their production could have cascading effects on other aspects of atmospheric chemistry. While gas-phase $RONO_2$ have been previously modeled above a forest canopy layer (Giacopelli et al., 2005; Pratt et al., 2012), the effect of physical and chemical differences above and below the forest canopy on their formation and their contribution to SOA has yet to be specifically investigated.

In addition to $RONO_2$ chemistry, recent work has highlighted pathways for SOA formation other than reversible partitioning of gas-phase oxidation products that may be also impacted by light and chemical gradients produced by forest canopies. For example, $O_3$ oxidation of monoterpenes has been observed to produce extremely low-volatility oxidation products that condense irreversibly onto existing aerosol, and these products may be a dominant relative contributor to SOA at low total mass loadings (Ehn et al., 2014; Jokinen et al., 2015). In addition, reactive uptake of isoprene epoxydiols, glyoxal, and methylglyoxal onto aerosol surfaces to produce non-volatile SOA has been shown to constitute a substantial fraction of total SOA mass-loadings in isoprene-dominated environments (Li et al., 2015).

Numerous studies have utilized one-dimensional models to investigate BVOC and radical chemistry within and above forests (Fuentes et al., 2006; Wolfe et al., 2011; Pratt et al., 2012; Rinne et al., 2012; Mogensen et al., 2015); however, to our knowledge only one study has used such a model to investigate SOA production (Ashworth et al., 2015), and the influence of chemical and physical differences above and below the forest canopy were not explicitly investigated. In addition, it was noted that SOA mass loadings were under-predicted within the canopy space. Furthermore, the majority of forest-atmosphere exchange models have considered relatively remote locations, encouraging an evaluation of the sensitivity of modeled results, especially with respect to the influence of $NO_3$ chemistry, to elevated background pollutant concentrations that may exist in suburban (or even urban), forested locations.

Understanding differences in atmospheric chemistry above and below the forest canopy is important for two reasons. First, as people living within or nearby forested environments breathe air under the canopy, substantial differences in mass loadings above and below the forest may lead to incorrect conclusions in regional air quality models on the outer edges of suburban metropolitan areas. Second, while vertical transport is suppressed within the forest layer relative to conditions without trees, exchange between the sub-canopy environment and the atmospheric boundary layer does occur, suggesting that products formed in the low-light conditions below the canopy affect above-canopy chemical processes and aerosol formation once transported, a phenomenon that may be particularly relevant for $NO_3$-derived $RONO_2$. To investigate these possibilities, we applied two coupled zero-dimensional (0D) models describing BVOC-$NO_3$-SOA chemistry with highly detailed chemical mechanisms to observations made above and below a forest canopy in a rural, deciduous forest environment during the intensive Community Atmosphere-Biosphere Interactions Experiments (CABINEX) 2009 field campaign.

## 2. Methods

### 2.1 Description of Model

The modeling structure used in this study incorporates the coupling of two 0D models designed to investigate detailed differences in chemical processing and SOA formation above and below the forest canopy layer. MATLAB computing software (v. R2014a) was used to perform a diel analysis of each environment (above and below the canopy) with output every second. The model uses a variable-order ordinary differential equation solver (MATLAB® ODE solver ode15s) to solve the underlying system of differential equations. One day of model spin-up was used for each analysis (Pratt et al., 2012). The oxidation mechanisms of α-pinene and isoprene were obtained directly from the Master Chemical Mechanism (MCM v3.2, via website: http://mcm.leeds.ac.uk/MCM, Jenkin et al., 1997; Saunders et al., 2003), as well as the standard gas-phase inorganic reactions included in the MCM, resulting in a system of 2260 chemical reactions with 712 chemical species. These two BVOCs were chosen because of their relatively high emission rates as well as differences in their reactivity and SOA formation potential. For instance, isoprene is known to react predominately with OH, while α-pinene reacts substantially with all three major oxidants within forests (Fuentes et al., 2007). The SOA yields measured during low-NO$_x$ OH oxidation of isoprene are generally lower than those from α-pinene; however, their SOA yields from NO$_3$ oxidation are similar (Table 1).

Chemical species within each 0D box (above and below the canopy) react in the gas phase via mechanisms dictated by the MCM, partition between the gas- and aerosol-phases if of sufficiently low volatility, undergo dry deposition, and are transported between environments. While the gas-phase chemical mechanism has been briefly introduced above, the remaining model processes are described in the sections below.

## 2.2 Modeling of Aerosol Processes

### 2.2.1 Equilibrium Gas-Particle Partitioning

The equilibrium gas-particle partitioning model developed by Colville and Griffin (2004) was used to quantify SOA production from individual isoprene and α-pinene oxidation products. Assuming that OA exists as a liquid, an equilibrium partitioning coefficient ($K_{om,i}$, m$^3$ μg$^{-1}$) for each oxidation product can be expressed as (Pankow 1994a,b; Odum et al., 1996)

$$K_{om,i} = \frac{A_i}{G_i M_o} = \frac{RT}{MW_{om} 10^6 \alpha_i p_{L,i}^o} \tag{1}$$

where $A_i$ is the mass concentrations of species $i$ in the aerosol phase (μg m$^{-3}$), $G_i$ is the mass concentration of species $i$ in the gas phase (μg m$^{-3}$), $M_o$ is the total mass concentration of the absorbing phase (μg m$^{-3}$), $R$ is the ideal gas constant (8.206 x 10$^{-5}$ m$^3$ atm mol$^{-1}$ K$^{-1}$), $T$ is the temperature (K), $MW_{om}$ is the average molecular weight of the absorbing phase (g mol$^{-1}$), $\alpha_i$ is the activity coefficient of oxidation product $i$ in the aerosol phase (here assumed ideal, $\alpha_i = 1$), and $p_{L,i}^o$ is the temperature dependent sub-cooled liquid vapor pressure of the species (atm). The sub-cooled liquid vapor pressure of each oxidation product was determined using the SIMPOL.1 group contribution method (Pankow and Asher, 2008).

Considering this definition in conjunction with the total species concentration (μg m⁻³), $C_i$ (based on the chemical mechanism described above) and a mass balance for the phase distribution:

$$\sum_{i=1}^{N} \frac{K_{om,i}C_i}{1+K_{om,i}M_o} + \frac{POA}{M_o} - 1 = 0 \qquad (2)$$

where POA represents any initially present OA. Using $C_i$ values determined by the gas-phase mechanism, this equation is solved for $M_o$ at every model time step, from which the value of $A_i$ for each of the $N$ species can be determined. It should be noted that this partitioning model fails to account for any effects related to the liquid water content (LWC) of SOA or any particle-phase reactions, which, in the case of non-oxidative accretion reactions, have the potential to significantly lower the volatility of the resulting SOA (Kroll and Seinfeld, 2008). As these particle-phase reactions can rapidly produce high MW, low volatility compounds, often more aerosol mass is produced in the environment than equilibrium partitioning of the gas-phase species would predict (Johnson et al., 2005; Kroll and Seinfeld, 2008).

Furthermore, as increases in RH, and subsequent increases in the water content of SOA, are known to enhance the partitioning of organic species to the aerosol phase, the omission of this process also potentially leads to underestimation of the total amount of SOA formed (Hennigan et al., 2008). However, as water uptake is generally driven by inorganic aerosol components (Hennigan et al., 2008; Carlton and Turpin, 2013), which only comprised a small fraction of total aerosol mass during CABINEX, the overall effect of RH is predicted to be small for the conditions of this study (VanReken et al., 2015).

Therefore, the primary uncertainty in aerosol formation is related to the production of high MW compounds through particle-phase reactions, but these compounds are partially accounted for through the parameterization of extremely low-volatility organic compound (ELVOC) formation described below. Following Ashworth et al. (2015), our model utilizes a POA value of 0.5 μg m⁻³ for calculation of SOA partitioning.

**2.2.2 Production of ELVOCs**

Recent studies have indicated that ELVOCs contribute substantially to monoterpene SOA mass loadings (Ehn et al., 2014; Jokinen et al., 2015). As a result, despite the MCM being the most complete chemical mechanism available, sole modeling of the partitioning of MCM oxidation products may result in an under-prediction of SOA formation. After initial oxidation by $O_3$ (or OH to a lesser extent), monoterpene peroxy radicals have the potential to undergo sequential H-shifts and $O_2$ additions that very rapidly produce ELVOCs with multiple hydroperoxide moieties (Jokinen et al., 2015; Mentel et al., 2015). Isoprene is also known to produce ELVOCs; however, measured yields are much smaller than those of monoterpenes (Jokinen et al., 2015). In SOA formation experiments with low total aerosol mass loadings, a characteristic similar to the relatively pristine environment near the CABINEX site used in this study, irreversible condensation of these ELVOCs represents a dominant fraction of total aerosol mass (Ehn et al., 2014).

A gas-phase ELVOC formation mechanism is included within the model based on observed molar ELVOC yields from α-pinene + $O_3$ (3.4%), α-pinene + OH (0.44%), isoprene + $O_3$ (0.01%), and isoprene + OH (0.04%) (Jokinen et al., 2015). Laboratory experiments suggest that in general, a distribution of

monomer ($C_{10}$) and dimer ($C_{19-20}$) ELVOC oxidation products contribute to SOA mass from α-pinene oxidation (Ehn et al., 2014), while isoprene is thought to form primarily monomer ($C_5$) species (Jokinen et al., 2015); however, as monomers are generally observed with a mass spectral signal an order of magnitude higher than those of dimers, and in order to ensure that uncertainty in the modeling parameters results in under-prediction rather than over-prediction of ELVOC mass loadings, all ELVOC products in the model presented here are assumed to be monomers. Specifically, α-pinene is assumed to produce the species $C_{10}H_{16}O_9$ while isoprene is assumed to produce $C_5H_{10}O_8$. At a representative temperature of $20^oC$, these products have saturation vapor pressures of $1.03 \times 10^{-14}$ atm and $2.81 \times 10^{-11}$ atm, respectively, according to the SIMPOL.1 method. These specific products were selected based on their intensity in observed ELVOC mass spectra and the fact that their elemental ratios are generally representative of the average ELVOC product distributions observed (Ehn et al., 2014; Jokinen et al., 2015).

### 2.2.3 Reactive Uptake of Isoprene Epoxides, Glyoxal, and Methylglyoxal

Another mechanism for aerosol formation known to be substantial in isoprene-dominated environments is the reactive uptake of isoprene epoxides, glyoxal, and methylglyoxal (Volkamer et al., 2007; Paulot et al., 2009; Li et al., 2015). Rather than reversibly partitioning between the gas and aerosol-phases, these specific oxidation products are irreversibly incorporated into SOA with rates controlled by particle acidity and the surface area of ambient OA. Isoprene epoxides (IEPOX) are formed through OH oxidation of isoprene hydroxyhydroperoxides and constitute a major fraction of SOA from isoprene oxidation in low-$NO_x$ conditions (Paulot et al., 2009; Lin et al., 2012). Glyoxal (CHOCHO) and methylglyoxal ($CH_3C(O)CHO$), the smallest dicarbonyl compounds, are formed from the oxidation of a variety of VOCs, and while they are known have short atmospheric lifetimes due to photolysis and OH oxidation, their high water solubility results in substantial uptake onto aqueous surfaces (Fu et al., 2009). After being incorporated into the aerosol, aqueous phase oxidation of these species results in the formation of non-volatile organic acids or oligomers, preventing partitioning back into the gas-phase (Loeffler et al., 2006; Carlton et al., 2007; Lin et al., 2012). Previous measurements in Mexico City reported significantly lower concentrations of glyoxal and methylglyoxal than gas-phase processes would predict, supporting the hypothesis of their reactive uptake into the particle phase (Volkamer et al., 2007). Isoprene epoxides have been identified in aerosol measurements at multiple locations (Chan et al., 2010; Froyd et al., 2010).

Following the method of Li et al. (2015), the surface-controlled uptake of these products can be expressed by:

$$\frac{dM_{air}}{dt} = -\left(\frac{1}{4}\gamma_i \nu_i A\right) G_i \tag{3}$$

where $\gamma_i$ is the reactive uptake coefficient (not the activity coefficient used in Eq. 1), $\nu_i$ is the thermal velocity of the gas species (m s$^{-1}$), $A$ is the ambient aerosol surface area concentration (m$^2$ m$^{-3}$), and $G_i$ is the mass concentration of the species in the gas-phase (μg m$^{-3}$) (the same quantity as $G_i$ in Eq. 1) . As the model does not explicitly calculate aerosol sizes, aerosol surface area concentrations were obtained from the study of VanReken et al. (2015) during CABINEX 2009. The reactive uptake coefficient represents the

probability that any collision between a gas-phase molecule and the aerosol surface will result in irreversible uptake into the aerosol phase. Glyoxal and methylglyoxal were assigned a reactive uptake coefficient of 2.9 x 10[-3], following the method of Fu et al. (2009). As this value is not adjusted for the LWC of aerosol measured at CABINEX, which is predicted to be low, our model may over-predict uptake of glyoxal and methylglyoxal to some extent; however, modeled mass loadings of both glyoxal and methylglyoxal SOA agree with simulations by Li et al. (2015) for northern Michigan. The reactive uptake coefficient of isoprene epoxides depends strongly on aerosol acidity, causing the value to vary substantially depending on the composition of the ambient aerosol (Wang et al., 2012). Using the Community Multi-Scale Air Quality (CMAQ) model, Li et al. (2015) calculated isoprene epoxide uptake coefficients based on particle acidity across the entire U.S. and found a value of around 0.5 x 10[-3] appropriate for conditions in northern Michigan. We utilized this value for this study.

### 2.2.4 Heterogeneous Hydrolysis of Organic Nitrates

Organic nitrates have a large influence on atmospheric chemistry through their impacts on $NO_x$, $O_3$ and OA. It has been shown that $RONO_2$ within the aerosol phase can be efficiently hydrolyzed, leading to the production of $HNO_3$ and the removal of $NO_x$ from the atmosphere (Day et al., 2010; Browne et al., 2013; Rindelaub et al., 2015; Bean and Hildebrandt Ruiz, 2016). Both field studies and chamber experiments have observed decreases in the $RONO_2$ content of SOA under conditions of increased relative humidity (Day et al., 2010; Bean and Hildebrandt Ruiz, 2016).

While many questions remain regarding the kinetics of the hydrolysis mechanism and the specific relationship with RH, chamber experiments have defined lifetimes of both isoprene and α-pinene nitrates within OA (Cole-Filipiak et al., 2010; Darer et al., 2011; Hu et al., 2011; Rindelaub et al., 2015; Bean and Hildebrandt Ruiz, 2016). For our study, both isoprene and α-pinene nitrate oxidation products are assumed to undergo first-order loss within the aerosol phase. Isoprene nitrate loss is parameterized using the average lifetimes found by Hu et al. (2011) for primary, secondary, and tertiary nitrate species. As primary and secondary nitrates only slowly hydrolyze even under the most acidic conditions observed (with a lifetime of ~500 hours at a pH of 0), which were not likely to occur during CABINEX, those products have an effective hydrolysis rate of zero within the model (Hu et al., 2011). However, tertiary nitrates, which are efficiently hydrolyzed even at neutral pH, have an effective lifetime of 0.67 hours within the aerosol (Hu et al., 2011).

To our knowledge, only two chamber experiments have been performed regarding the hydrolysis of α-pinene nitrate oxidation products within OA (Rindelaub et al., 2015; Bean and Hildebrandt-Ruiz, 2016). Of these, only the study by Bean and Hildebrandt Ruiz (2016) quantified hydrolysis rates. The overall hydrolysis rate was found to depend strongly on RH. For instance, experiments with an RH between 20 and 60% produced a hydrolysis rate of 2 day[-1], while those with RH above 70% had rates around 7 day[-1] (Bean and Hildebrandt-Ruiz, 2016). As RH values measured during CABINEX were generally above 60%, our model uses a rate of 7 day[-1], corresponding to a lifetime of 3.4 hours.

**2.3 Deposition**

Dry deposition of gases and aerosols is included within both the above- and below-canopy boxes. In the above-canopy model, deposition is assumed to occur onto the top of the canopy, and the resistance to deposition is determined using the method of Meyers and Baldocchi (1988). Deposition velocities for each chemical species are calculated based on resistances from the quasi-laminar boundary layer and leaf mesophyll, cuticular surfaces, and stomata. A thorough description of the particular equations used in this method can be found in Bryan et al. (2012). The specific parameter values used in the calculation of deposition velocities are listed in Table S1. The above-canopy box height was assumed to vary diurnally based on the boundary layer values utilized in Giacopelli et al. (2005) for a previous model of the CABINEX location. As our model does not calculate aerosol sizes, size distribution data obtained from VanReken et al. (2015) were used to calculate a volume-weighted average settling velocity of aerosol particles.

In the below-canopy model, deposition was assumed to occur to the ground and was modeled following the method of Gao et al. (1993). This method assumes that the deposition velocity is based on the combination of aerodynamic resistance near the ground and a species-specific resistance to deposition to the forest floor. The box height was set to 6m (the assumed bottom of the canopy layer or top of the trunk space) for the entire diurnal period.

**2.4 Vertical Transport**

Recent modeling studies have revealed that in-canopy vertical mixing substantially influences overall forest chemistry (Bryan et al., 2012). Mixing is often parameterized in one-dimensional (1D) models using traditional K-theory, which quantifies the transport of chemical species by eddy diffusion at a rate corresponding to the heat exchange coefficient (Wolfe et al., 2011; Bryan et al., 2012). While one of the most accurate turbulence parameterization methods available, such a process is computationally sophisticated and is better suited to 1D modeling of an environment. In order to simplify the quantification of mixing, we have implemented the concept of in-canopy residence lifetimes to determine mixing rates. Measured and modeled in-canopy residence times vary substantially depending on the forest environment studied. For instance, Fuentes et al. (2007) report average residence times of ~8 minutes for a parcel emitted near the ground, during the day, in a forest with a 26-m high canopy, while Farmer and Cohen (2008) calculate residence times of 1-7 minutes for a forest with a canopy height of only 5.7m. Maximum residence times of up to 50 minutes have been reported in tall forests (Strong et al., 2004). Transport back into the canopy is an even more complicated process, as coherent structures (i.e., sweeps of air downward), rather than simple turbulence, often produce the majority of scalar fluxes (Steiner et al., 2011).

For our model, a minimum residence time of ten minutes in each location is assumed to apply midday, in agreement with previous measurements, and residence times vary diurnally based on the modeled turbulent heat exchange coefficient produced by the CACHE model (Bryan et al., 2012). In

addition, we have included a diurnal above-canopy vertical dilution rate based on the average of methanol and acetaldehyde above-canopy vertical loss calculated by the FORCAsT 1-D model for summertime conditions (Ashworth et al., 2016).

The inclusion of a mixing process, vertical dilution, and dry deposition results in relatively good agreement between measured and modeled concentrations of the sum of the isoprene oxidation products methacrolein and methylvinyl ketone (MACR+MVK) (Fig. S1). These photooxidation products of isoprene are often significantly overestimated in low $NO_x$ conditions modeled in forests (Bryan et al., 2012; Ashworth et al., 2015). The validity of the mixing process specifically is supported by the agreement between the measured and modeled differences in MACR+MVK mixing ratios above and below the canopy (Fig. S1c), as this comparison removes the influence of errors in the chemical mechanism that may produce over- or under-predictions of the overall mixing ratios. While the comparison of differences in mixing ratios above and below the canopy indicates that we likely over-predict mixing in the early morning and under-predict mixing in the late evening, this study is primarily focused on daytime conditions, during which the differences are within a few percent. Different minimum transport lifetimes above and below the canopy (from 10 minutes to 30 minutes) were analyzed to determine if better agreement between measured and modeled MACR+MVK profiles could be attained; however, this ultimately produced little variation in the overall agreement, and the use of a ten minute lifetime in each location was found to produce minimum error between measured and modeled MACR+MVK mixing ratios (Fig. S1d).

**2.5 Input Data**

The CABINEX campaign took place in northern Michigan near the University of Michigan Biological Station (UMBS) in August 2009 and utilized the Program on Oxidants: Photochemistry, Emissions, and Transport (PROPHET) tower to perform gas-phase measurements throughout the canopy. A deciduous forest with a canopy height of around 22.5 m surrounds the PROPHET tower and consists of tree species that emit significant quantities of both isoprene (aspen and oak) and monoterpenes (pine and birch). Above canopy measurements were made at a height of 34 m, while those below canopy occurred at a height of 6 m (Fig. S2). More detailed descriptions of both the PROPHET tower and the surrounding environment can be found in the literature (Carroll et al., 2001; Ortega et al., 2007; Griffith et al., 2013).

The primary goal of the campaign was to examine the effect of forest succession on atmospheric chemistry. The majority of studies resulting from the CABINEX campaign have focused on improving understanding of the linkage between gas-phase radical and BVOC chemistry through measurements and modeling (Kim et al., 2011; Bryan et al., 2012; Griffith et al., 2013; Hansen et al., 2014), while two have investigated aerosol concentrations at the site (Ashworth et al., 2015; VanReken et al., 2015). Aerosol size and composition vary widely based on ambient wind direction but show a stronger anthropogenic influence when air masses come from populated areas to the east and south (VanReken et al., 2015). Accurate modeling of diel changes in SOA mass loadings using a 1-D model has thus far proven difficult (Ashworth et al., 2015).

The model was constrained by diel median values of hydrogen oxides ($HO_x$ = OH + hydroperoxy radical ($HO_2$)), NO, $NO_2$ $O_3$, $\alpha$-pinene (monoterpenes), isoprene, formaldehyde (HCHO), and the photolysis rate of $NO_2$. Each of these species was therefore assigned a median measured value at every hour, and values used for each model time step (e.g. 8:03) were determined by linear interpolation between hourly values. The Indiana University Fluorescence by Gas Expansion (IU-FAGE) instrument was used to measure OH and $HO_2$ (Griffith et al., 2013). Interferences involved in the operation of the IU-FAGE instrument caused slight positive artifacts in the measurement of both OH and $HO_2$ concentrations. During below canopy measurements, laser photolysis of $O_3$ within the IU-FAGE sampling cell resulted in a minor artificial increase in OH concentrations, while both above and below the canopy, a fraction of peroxy radicals ($RO_2$) was converted into $HO_2$ within the sampling cell, as further explained in Section 2.3.1. The resulting influence of OH on BVOC chemistry below the canopy therefore represents an upper limit. Nitrogen oxides were measured using a two-channel chemiluminescense instrument with a blue-light converter for $NO_2$ measurements (Air Quality Design, Inc.), $O_3$ was measured using ultraviolet absorption (Thermo Environmental Instruments Inc. 49c), and isoprene, monoterpenes, and formaldehyde were measured with a proton-transfer reaction mass spectrometer (PTR-MS) (IONICON, Inc.). As the PTR-MS did not distinguish between monoterpene species, all monoterpenes were assumed to be $\alpha$-pinene for modeling purposes even though its $RONO_2$ and SOA yields are generally smaller than those of other monoterpene species (Fry et al., 2014; Zhao et al., 2015), likely resulting in a low bias. This assumption is justified by noting that gas chromatography-mass spectrometry (GC-MS) measurements taken at a height of 6m during CABINEX indicate $\alpha$-pinene accounted for an average of ~77% of monoterpenes at the site (Wallace, 2013). The photolysis frequency of $NO_2$ was measured with a Scanning Actinic Flux Spectroradiometer (Flynn et al., 2010). Further information regarding the measurement techniques can be found in Griffith et al. (2013).

Above canopy photolysis frequencies for isoprene and $\alpha$-pinene oxidation products were obtained directly from the MCM, while photolysis frequencies for other species were taken from the National Center for Atmospheric Research (NCAR) Tropospheric Ultraviolet and Visible (TUV) Model (TUV Model 4.1, via website: http://cprm.acom.ucar.edu/Models/TUV/Interactive_TUV/). In order to correct for non-clear sky conditions, above canopy photolysis frequencies were scaled to represent differences between the measured $NO_2$ photolysis frequency and that predicted by the TUV model. Below canopy frequencies were then calculated by scaling by the ratio of the below-to-above canopy $NO_2$ photolysis frequencies measured during CABINEX. This ratio is time-of-day dependent, with a maximum around noon, and varies over the range of 0.05 to 0.17 during daylight hours (Fig. S3d). Model input data are shown in the in the SI (Fig. S3).

At CABINEX, $O_3$ levels were relatively consistent throughout the day, reaching a maximum in the afternoon (Fig. S3). Median mixing ratios were consistently ~5 to 10 ppbv larger above the canopy than below (~30-35 ppbv above, ~20-30 ppbv below). While $NO_2$ concentrations peaked at night (~1 ppbv), likely due to both $O_3$ oxidation of local NO and transport from non-local air masses, they were larger below

the canopy in the early morning.  Concentrations of NO were below the detection limit of the instrument (6.7 pptv) for much of the night and were therefore held at 6.7 pptv for modeling purposes during these periods.  Maximum daytime NO mixing ratios reached ~0.2 ppbv in the early morning, largely the result of $NO_2$ photolysis (Seok et al., 2013). Monoterpene ($\alpha$-pinene) concentrations were consistent at around 0.3 ppbv, although they exhibited slight diel variability with maxima occurring at night, whereas the isoprene profile displayed strong emission dependence on sunlight and temperature and reached a maximum mixing ratio of ~1.6-1.7 ppbv mid-afternoon.

**2.6 Modification of gas-phase $RONO_2$ chemical mechanism**

Without any changes to the MCM, predicted gas-phase $RONO_2$ concentrations (~400 pptv) are much larger than those modeled by Pratt et al. (2012) (~20-70 pptv), implying that gas-phase $RONO_2$ is over-predicted by the MCM. Dry deposition rates of both isoprene and $\alpha$-pinene nitrates (~1.5-2.5 cm s$^{-1}$) are similar to the values used in Pratt et al. (2012), and the vertical dilution rate used in our model is comparable to the value used by Giacopelli et al. (2005) for nitrates at UMBS, suggesting that physical processes are not the reason for the modeled difference.

The discrepancy in total $RONO_2$ concentrations therefore likely stems from inappropriately large $RONO_2$ yields or inappropriately small loss rates within the MCM. While monoterpene nitrate concentrations modeled in previous studies are generally less than 50 pptv (Pratt et al., 2012; Fisher et al., 2016), without applying any changes to the chemical mechanism obtained from the MCM, two first-generation $\alpha$-pinene hydro-hydroxynitrates, produced from $NO_3$ oxidation followed by $RO_2$ + $HO_2$ reaction, have a cumulative concentration of over 100 pptv for the majority of the day (Fig. S4a). Within the MCM, the first-generation $RO_2$ + $HO_2$ reaction that forms these products has a 100% hydro-hydroxynitrate yield; however, laboratory experiments have indicated that while such a yield may be suitable for small peroxy radicals such as methyl peroxy, larger peroxy radicals are more likely to produce OH or alcohols upon reaction with $HO_2$ (Jenkin et al., 2007; Crowley and Dillon, 2008). Recent modeling by Fisher et al. (2016) using the GEOS-Chem model assumes that the first generation monoterpene $RO_2$ from $NO_3$ oxidation has only a 10% probability of nitrate retention upon reaction. In order to account for the laboratory findings and align our chemical scheme with that of Fisher et al. (2016), we reduced the production rate of the two hydro-hydroxynitrate compounds by a factor of ten. This results in an average nitrate retention probability of 8.5% above-canopy and 12.8% below-canopy for $\alpha$-pinene $RO_2$ from $NO_3$ oxidation. The finalized concentrations, while still larger than those modeled by Pratt et al. (2012), show much better agreement with previous findings.

Second-generation isoprene nitrate concentrations (~100 pptv) also appear to be over-predicted relative to previous observations, which report concentrations of 20-50 pptv (Pratt et al., 2012; Fisher et al., 2016). Similarly to the first-generation $\alpha$-pinene nitrate concentrations, modeled second-generation isoprene nitrate mixing ratios are dominated by two species, methyl vinyl ketone nitrate (MVKN) and methacrolein nitrate (MACRN). The MCM does not assume these products react with $NO_3$ or $O_3$, as has

been recently suggested (Kerdouci et al., 2010), and the reaction rate of each species with OH is 5-10 times larger in the model used by Pratt et al. (2012) than in the MCM. Adding reactions with $NO_3$ and $O_3$ and modifying the reaction rate with OH to match that used in Pratt et al. (2012) improves agreement (Fig. S4); however, the implications of the remaining differences are discussed in Section 3.3. Ultimately, as will be discussed, the $RONO_2$ fraction of SOA agrees well with previous measurements, indicating that the possible remaining errors in gas-phase $RONO_2$ concentrations should not unduly influence the conclusions related to SOA mass loadings.

## 2.7 Model Evaluation

Measured OH and $HO_2$ profiles above the canopy were compared to those predicted by our model when leaving $HO_x$ concentrations unconstrained, which gives an indication of the validity of both the modeled oxidation processes and photolysis frequencies, as both have a large effect on $HO_x$ chemistry (Fig. S5). The model tends to under predict nighttime OH concentrations, similarly to previous model comparisons (Carslaw et al., 2001; Ren et al., 2006; Pugh et al., 2010). On average, nighttime (22:00-6:00 local) modeled OH concentrations are ~$2.5 \times 10^5$ mol cm$^{-3}$ below those observed (~60% difference). As the model only incorporates $\alpha$-pinene (for all monoterpenes) and isoprene, the absence of OH-producing ozonolysis reactions with other VOCs likely contributes to this nighttime discrepancy. These reactions have been shown to produce up to ~64-72% of nighttime OH in rural environments (Bey et al., 1997; Geyer et al., 2003b). Daytime concentrations are also generally under-predicted, especially during the late morning when the measured OH concentration peaks. Despite the observed difference in the two profiles, only two measured points (11:00, 19:00) fail to capture modeled concentrations within their 68% confidence intervals. A linear regression of measured versus modeled OH concentrations highlights the ability of the model to capture the diel trend of OH concentrations ($r^2$ ~0.82) (Fig. S6).

While modeling studies have generally found better performance for $HO_2$ than for OH, many models involving $HO_x$ cycling in forested environments tend to underestimate $HO_2$ levels (Lelieveld et al., 2008; Pugh et al., 2010; Stavrakou et al., 2010). Interferences involved in the operation of the IU-FAGE instrument can lead to conversion of isoprene-derived peroxy radicals into $HO_2$, resulting in a positive artifact (Fuchs et al., 2011). Tests indicate that ~90% of isoprene-based hydroxyalkylperoxy radicals are converted into $HO_2$ in the sampling cell through reaction with NO and subsequent decomposition (Griffith et al., 2013). As a result, IU-FAGE measurements represent both ambient $HO_2$ and a fraction of isoprene peroxy radicals chemically converted to $HO_2$ within the instrument itself. Griffith et al. (2013) noted that $HO_2$ concentrations measured during CABINEX are similar to previously reported concentrations of $HO_2$ + $RO_2$ (Mihele and Hastie, 2003). To account for this, simulated isoprene-peroxy radical concentrations were added to simulated $HO_2$ concentrations when performing a regression analysis, and the discussion of the agreement between measurements and model output focuses on simulated $HO_2$ + isoprene $RO_2$ rather than simply $HO_2$.

While the model results successfully represent the overall diel $HO_2$ profile and show good agreement at night, modeled concentrations of $HO_2$ + isoprene $RO_2$ overestimate measured concentrations midday by about 30%, suggesting either a missing isoprene $RO_2$ loss mechanism, $HO_2$ loss mechanism, or a combination of both. Nevertheless, good agreement is observed between measured and modeled concentrations with and without the addition of isoprene $RO_2$ ($r^2 = 0.94$, slope = 0.68 with $RO_2$, $r^2 = 0.9$, slope = 1.2 without). The observation that adding isoprene $RO_2$ improves the coefficient of determination supports the notion of an isoprene-derived interference (Fig. S6). Ultimately, the agreement observed between measured and modeled $HO_x$ concentrations relative to previously published atmospheric chemistry models supports the validity of the modeling process.

## 3. Results and Discussion

### 3.1 Nitrate Radical Concentrations

Air surrounding the PROPHET tower is free from major $NO_x$ sources, causing soil emissions and long-range transport to be dominant influences on $NO_x$ mixing ratios (Alaghmand et al., 2012; Seok et al., 2013). This ultimately leads to low $NO_2$ (<1 ppbv) (Fig. S3) and correspondingly low predicted $NO_3$ concentrations (<1 pptv) (Fig. 1a), despite a relatively high ratio of $NO_2$ to NO (median ~12.3). Mixing ratios under 1 pptv agree well with previous results from Mogensen et al. (2015) in a boreal forest setting and Ayres et al. (2015) in rural Alabama. While this concentration is much lower than previous nighttime observations in polluted urban environments (Brown et al. 2011), model results have shown that $NO_3$ can dominate the total nighttime oxidative strength (defined as reactivity multiplied by concentration) of a forest environment, even at concentrations of less than 1 pptv (Mogensen et al. 2015).

The above-canopy ambient $NO_3$ profile also agrees well with modeled $NO_3$ concentrations at the same site for conditions in 2008 (Pratt et al., 2012). The model of Pratt et al. (2012) includes more BVOCs but less detail in terms of subsequent oxidation chemistry, and the results do not extend below the canopy, preventing a comparison. At night, $NO_3$ concentrations are enhanced above the canopy compared to below, largely due to larger $NO_2$ concentrations measured above the canopy during that period. During the day this trend reverses, and mixing ratios are on average three times as large below the canopy as above (0.07 pptv below and 0.023 pptv above). As $NO_2$ concentrations were relatively similar in both locations during the day and $O_3$ was elevated above the canopy, this observation points to a substantial difference in $NO_3$ loss rates in the two locations. The effect of $NO_3$ photolysis specifically is highlighted by the observation of a much stronger diel trend in concentrations above-canopy than below.

The rural nature of the UMBS site prevents large $NO_3$ mixing ratios, so in order to test the response of this environment to increased background pollution, we performed a simple sensitivity analysis wherein the diurnal profiles of $O_3$ and NO were scaled from their original values up to factors of 2 and 5 respectively. In addition, the constraints on OH, $HO_2$, and $NO_2$ were removed, in order to allow accurate modeling of both $HO_x$ concentrations and the NO to $NO_2$ ratio. The most polluted case studied, where $O_3$

mixing ratios are doubled and NO mixing ratios are multiplied by a factor of 5, produces a maximum $NO_x$ mixing ratio of ~10 ppb and is meant to represent a hypothetical suburban area ideally suited for $NO_3$ formation (Fig. S7). In this most extreme case, $NO_3$ concentrations increase by a factor of 10 to 12 relative to the conditions observed during CABINEX, resulting in a 0.4 pptv difference on average between the above-canopy and below-canopy environments from 11:00 to 15:00 (Fig. 1b). The implications of this increased difference are investigated subsequently.

Production and loss rates of $NO_3$ were analyzed to determine the specific reason(s) for elevated below-canopy concentrations during the day under the CABINEX conditions. From 6:00-12:00, the modeled enhancement of $NO_3$ below the canopy is primarily the result of faster production (due to higher $NO_2$ mixing ratios below than above the canopy), as loss rates are similar (Fig. 2). However, from 12:00-18:00 production rates are similar in both environments, and loss rates from NO and VOCs are higher below-canopy than above, indicating that reduced below-canopy $NO_3$ photolysis is the primary contributor to the observation of elevated below-canopy $NO_3$ concentrations during the afternoon.

**3.2 BVOC Oxidation**

Numerous studies have investigated $NO_3$ oxidation of BVOCs and highlighted that such oxidation is especially important at night (Golz et al., 2001; Brown et al., 2005; Brown et al., 2011; Stutz et al., 2010; Brown and Stutz, 2012). Certain classes of BVOCs also show appreciable daytime oxidation rates by $NO_3$ (Geyer et al., 2003a; Brown et al., 2005). The prediction of elevated daytime $NO_3$ concentrations below the forest canopy implies an increased rate of BVOC oxidation in that environment. Oxidation rates (and fractional contributions of the total) by each oxidant are determined by

$$\text{Oxidation Rate (ppbv hr}^{-1}\text{)} = k \text{ [Oxidant][VOC]} \qquad (5)$$

Ambient CABINEX results (Figs. 3 and 4) show that overall rates of $\alpha$-pinene oxidation above and below the canopy are similar (~0.06-0.1 ppbv hr$^{-1}$), while the fractional plots indicate that in both cases, $O_3$ is the most consistent oxidant, contributing over 40% of total oxidation. Hydroxyl radical oxidation reaches maximum contributions of 67% above the canopy and 62% below the canopy near midday (Fig. 4). The $NO_3$ oxidation rates largely reflect the $NO_3$ concentration profile, as $\alpha$-pinene concentrations are relatively consistent throughout the 24-hour period, with above-canopy rates displaying more diel variation than those below, due to the substantial effect of $NO_3$ photolysis. Oxidation by $NO_3$ contributes 17% of the total above the canopy on average and 18% below. This similarity in averages is due to the fact that daytime $NO_3$ concentrations are elevated below the canopy relative to above, whereas at night the opposite occurs. The most substantial difference between the two environments occurs from around 8:00 to 20:00, as $NO_3$ contributes on average only 4% of total $\alpha$-pinene oxidation above the canopy but as much as 12% below.

Increasing the concentration of background pollutants to simulate a suburban environment raises the fractional amount of $NO_3$ oxidation to as high as 32% during the day (8:00-20:00) below the canopy, similar to the model results of Fuentes et al. (2007) for a forest with 40-60 ppbv of $O_3$ and maximum $NO_x$

mixing ratios of 6-8 ppbv. Even during midday when $NO_3$ concentrations are minimized, the contribution of $NO_3$ to total α-pinene oxidation below the canopy is ~18% in this simulated polluted environment (Fig. 5).

Isoprene oxidation is only briefly discussed, as the total rates and fractional oxidant contributions are similar above and below the canopy (Figs. 3 and 4). In both locations, OH dominates isoprene oxidation, especially during the day when oxidation rates are most substantial. Peak below-canopy oxidation rates are generally smaller than those above, as expected, largely due to increased photolytic activity and OH concentrations above the canopy. Even in the highly polluted conditions, $NO_3$ oxidation accounts for only ~3% of the total near midday (Fig. 5).

### 3.3 Gas-Phase $RONO_2$ Concentrations

The diurnal profile of the sum of isoprene- and α-pinene-derived gas-phase $RONO_2$ is shown in Figure 6 for both above- and below-canopy environments. Isoprene is the major precursor of total $RONO_2$ (~70-85%), following from its larger mixing ratios and much faster daytime oxidation rates than α-pinene. This is in agreement with other isoprene-dominated environments (Pratt et al., 2012; Fisher et al., 2016; Romer et al., 2016). First generation nitrates produced from $NO_3$ oxidation of both BVOCs have much larger concentrations than the comparable second generation products, which have virtually negligible concentrations, implying that further oxidation tends to remove the nitrate functional group, after which point further reaction with NO or $NO_3$ and regeneration of the nitrate group is unlikely, as suggested by Lee et al. (2014) and Xiong et al. (2016). As a result, despite the fact that OH only contributes around 20% of nighttime α-pinene oxidation, $RONO_2$ products formed from OH oxidation of α-pinene constitute over half of nighttime α-pinene $RONO_2$ concentrations.

The efficient mixing process parameterized within the model results in very similar overall concentrations above and below the canopy (within 9%), despite the modeled differences in BVOC oxidation. While this appears to imply that the canopy exerts little influence on $RONO_2$ production, midday (11:00-15:00) above-canopy mixing ratios of $NO_3$-derived $RONO_2$ products formed from α-pinene and isoprene are 15.2% and 33.7% larger on average than if the mixing process is turned off and the dilution rate is reduced by one-half (to partially account for the lack of oxidation product transport from below the canopy). Therefore, near midday, the below-canopy environment appears to act as a source of $NO_3$-derived nitrates above the canopy, a finding that is further explored subsequently. We use the term "$NO_3$-derived" extensively throughout the remainder of this manuscript to indicate products formed from initial oxidation of BVOCs by $NO_3$.

Overall $RONO_2$ accounts for ~7-12% of total $NO_y$ in both environments, which, while somewhat smaller than observations in the eastern U.S. by Perring et al. (2009) (~18%), nevertheless suggests that $RONO_2$ fate is highly relevant to overall $NO_x$ cycling within this environment. However, inconsistencies between studies of $RONO_2$ chemistry underscore the need for continued study of the oxidation pathways involved in $RONO_2$ production and agreement between published mechanisms and laboratory results. For

instance, despite our changes to the chemical mechanism, our MCM-based model predicts over twice as much total gas-phase $RONO_2$ as the model of Pratt et al. (2012). Furthermore, second-generation products constitute 50-60% of total $RONO_2$ (predominantly from the OH-NO pathway) in our model, while models based on laboratory yields, such as that of Pratt et al. (2012), predict almost entirely first-generation products. Observations of isoprene nitrates suggest second-generation products comprise somewhere between 20-50% of the total (Fisher et al., 2016).

In a rural area where $RONO_2$ formation is limited, mischaracterizing $RONO_2$ composition may have a negligible effect on predicting subsequent aspects of atmospheric chemistry; however, in more polluted areas both local and regional $O_3$ and SOA formation rates may be significantly affected by such an error, as $NO_x$ reservoirs such as $RONO_2$ can limit $O_3$ production, and first and second generation $RONO_2$ products are thought to have different deposition velocities, saturation vapor pressures, and tendencies to release $NO_x$ upon oxidation (Lee et al., 2014; Fisher et al., 2016).

### 3.4 $RONO_2$ Production, Loss, and Net Transport

While modeled $RONO_2$ mixing ratios above and below the canopy are similar due to efficient daytime mixing within the forest canopy, the physical and chemical differences described previously suggest that production and loss rates differ between locations. The diurnal profiles of simulated $RONO_2$ production rates above and below the canopy are shown in Figure 7. The total midday production rate of $RONO_2$ above the canopy (~100 pptv) and the dominance of isoprene-derived $RONO_2$ production agree with summertime results obtained for the Southeast U.S. in another deciduous forest (Romer et al., 2016).

In both locations, a large increase in production rates occurs around 6:00, corresponding to the rapid increase in OH, NO, and isoprene concentrations. Production of $RONO_2$ through $NO_3$ oxidation is faster below the canopy than above during the day; however, the difference, a factor 2.75 for α-pinene and 1.35 for isoprene, is slightly lower than the difference in oxidation rates. Following initial oxidation by $NO_3$, the produced $RO_2$ must react with $HO_2$ or $RO_2$ in order to form a stable nitrate, both of which have higher concentrations above the canopy than below (production through NO reaction after $NO_3$ oxidation is negligible). As a result, while daytime oxidation by $NO_3$ is much more rapid below the canopy, the differences in mixing ratios of species involved in secondary reactions slightly offset the difference in $NO_3$ concentrations. Overall, $NO_3$ accounts for ~10% of total $RONO_2$ production above the canopy and 17% below near midday (11:00-15:00).

In the more polluted environment, the fraction of total daytime $RONO_2$ produced by $NO_3$ oxidation increases to 14% above the canopy and 20-25% below the canopy (Fig. 8). Therefore, while $NO_3$ chemistry becomes more relevant during the day under more polluted conditions, the relative difference between above- and below-canopy production through $NO_3$ oxidation (6-11%) stays relatively constant.

Loss rates of both α-pinene and isoprene $RONO_2$ are shown in Figure 9. On average, photolysis and gas-phase oxidation, processes that tend to release $NO_x$, account for approximately half of α-pinene $RONO_2$ loss both above and below the canopy (41% above and 47% below) and an even higher fraction of

isoprene loss (65% above and 48% below), with loss due to photolysis displaying a strong difference between above and below canopy environments. Depositional losses are relatively consistent above and below the canopy, as a larger box height above the canopy (the boundary layer height) is offset by faster daytime mixing conditions (deposition velocities) above than below the canopy.

Modeled hydrolysis accounts for less than 5% of total $RONO_2$ loss on average, a much smaller amount than previous estimations (~30-50%) based on measurements (Romer et al., 2016) and model predictions (Fisher et al., 2016). However, a few important differences between our model and the assumptions of these previous studies contribute to this difference. The observational study by Romer et al. (2016) assumes no loss of $RONO_2$ to photolysis, which the MCM predicts is a substantial loss process for both α-pinene and isoprene nitrates near midday, while the modeling study of Fisher et al. (2016) assumes all isoprene organic nitrate species hydrolyze at the same rate as tertiary nitrates, likely leading to an over-prediction of hydrolysis rates for a given fraction of $RONO_2$ within the aerosol phase in their model.

While the difference between our results and previous measurements of Romer et al. (2016) would be reduced somewhat if photolysis were included in their analysis, their measurements nevertheless suggest a large fraction of $RONO_2$ is lost to hydrolysis even in low-light conditions (6:00-7:00), implying an under-prediction by our model of the fraction of $RONO_2$ mass in the aerosol phase. Previous studies using the MCM to predict SOA composition have had to reduce gas-particle partitioning coefficients by factors as high as 500 in order to reproduce observed mass loadings (Johnson et al., 2004; Johnson et al., 2006a, 2006b). Essentially this reduction accounts for particle-phase reactions (not captured by modeling simple semi-volatile partitioning) and particle viscosity that decrease the probability of repartitioning into the gas phase. Decreasing the vapor pressures of all BVOC oxidation products by a factor of 100 increases hydrolysis rates substantially, causing hydrolysis to account for a minimum of 40-50% of α-pinene $RONO_2$ loss above-canopy and 60-70% below-canopy (Fig. S8). The fraction of isoprene $RONO_2$ lost through this mechanism also increases, but only to a maximum of 10-15% at night, as only tertiary nitrates undergo hydrolysis. The uncertainty with respect to hydrolysis loss rates highlights the importance of more accurately characterizing gas-particle partitioning when quantifying the lifetime and fate of $NO_x$ within an environment.

As we are focused primarily on highlighting differences between the above- and below-canopy locations rather than precisely determining production and loss rates, this uncertainty does not affect our major conclusions regarding the increased relevance of $NO_3$ chemistry below the canopy. In addition, we are primarily focused on daytime conditions, when higher temperatures shift partitioning to the gas-phase and reduce overall hydrolysis rates. Nevertheless, future versions of this model will look to improve the characterization of hydrolysis by more accurately describing both hydrolysis rates within the aerosol (when more precise data are available) and more accurately partitioning the suite of BVOC oxidation products between the gas and particle phases. In addition, the choice of α-pinene to represent all monoterpenes can impact these findings due to changes in product distribution that would occur if other monoterpenes were considered.

When considering all relevant losses, midday $RONO_2$ loss rates below the canopy are less than one-half of those above the canopy for α-pinene products and are around 66% of those above the canopy for isoprene products. As daytime production rates of $NO_3$-derived $RONO_2$ below the canopy are larger than those above the canopy, our results therefore suggest possible net transport of $NO_3$-derived products upward from the below-canopy environment. The net transport of both total $RONO_2$ and $NO_3$-derived $RONO_2$ are shown in Figure 10.

While net transport of total $RONO_2$ is generally from the above-canopy environment to the below-canopy environment (downward), in agreement with previous $NO_y$ flux measurements at UMBS (Geddes and Murphy, 2014), the below-canopy environment acts as a substantial source of $NO_3$-derived $RONO_2$ above the canopy during the day (~25%). The largest contribution from net transport (~30% of the total) occurs around 7:00. Recent modeling work indicates that the mass loading of particulate $RONO_2$ is maximized around this time in the southeast U.S. (Pye et al., 2015). As $NO_3$-derived products are known to be major constituents of SOA in monoterpene-rich areas such as the southeast U.S., neglecting below-canopy chemistry could therefore lead to an under-prediction of $RONO_2$ SOA mass above the canopy during the day (Ayres et al., 2015; Xu et al., 2015).

While decreasing the vapor pressures of all oxidation products by a factor of 100 decreases the fraction of $RONO_2$ loss caused by photolysis and OH oxidation, the total contribution of net transport to above-canopy $NO_3$-derived $RONO_2$ production is still 14.5% on average from 6:00-18:00 (Fig. S9). Under more polluted conditions, the contribution of transport to total production remains substantial (between 7-25% from 11:00 and 15:00), but is highly dependent on the ratio of $O_3$ to NO (Fig. S10). Our assumption of no depositional losses during transport likely leads to an over-prediction of the total amount of net transport that occurs; however, our model under-predicts concentrations of MACR+MVK above and below the canopy during the morning when mixing is beginning (Fig. S1), suggesting that losses during transport may be captured in the parameterization of deposition. Such under-prediction could also be the result of errors in the chemical mechanism.

Precise determination of the influence of transport on above-canopy $NO_3$-derived $RONO_2$ production requires more accurate characterization of both hydrolysis losses and deposition during transport than our model can currently achieve. However, our results suggest that $NO_3$-derived products and total $RONO_2$ may undergo net transport in opposite directions. Because of the known relevance of $NO_3$-derived products on SOA production, this possibility should be the focus of further investigation.

**3.5 Secondary Organic Aerosol Production**

Figure 11 displays the simulated diurnal SOA mass loadings modeled above and below the canopy at UMBS. As with gas-phase $RONO_2$, efficient daytime mixing results in a relatively small difference in total SOA mass loading between locations, with an average difference of 6% at night (21:00-6:00) (higher mass loadings below-canopy) and 7% near midday (11:00-15:00) (higher mass loading above-canopy). The

similarity above and below the canopy agrees with the results of the 1-D FORCAsT model (Ashworth et al., 2015), which predicts little change in aerosol mass loadings within 100m from the ground.

To our knowledge, the composition of ambient aerosol at UMBS has only been characterized once before. Aerosol mass spectrometer measurements from 2001 indicate that OA mass loadings varied from ~0.5 to 2 μg m$^{-3}$, with values reaching 3 μg m$^{-3}$ during peak pollution events (Delia, 2004), in relatively good qualitative agreement with our results. However, more recent data from CABINEX 2009 indicate that water-soluble organic mass loadings were around 4 μg m$^{-3}$ on average (VanReken et al., 2015). As a result, while our model predicts more aerosol mass than recent results from the FORCAsT model (Ashworth et al., 2015), total SOA mass loadings may still be somewhat under-predicted.

In terms of SOA composition, ELVOCs comprise a consistent majority of total SOA in the model (~77% above-canopy and ~72% below-canopy), which follows from the nearly constant rate of O$_3$ oxidation of α-pinene. The dominance of these products agrees with laboratory studies that find that SOA composition is almost entirely composed of ELVOC species at low total SOA mass loadings (Ehn et al., 2014). However, as the molar yield of ELVOCs varies substantially based on monoterpene studied (Jokinen et al., 2015), and as ELVOCs are not included in major chemical mechanisms such as the MCM, the modeled relevance of this type of oxidation product to SOA mass loadings suggests future studies should look to improve characterization of ELVOC formation processes and update published mechanisms.

Similar amounts of SOA are produced by reactive surface uptake of isoprene products (IEPOX, GLYOX, and METHYLGLYOX) and semi-volatile oxidation product partitioning (ROOH and RONO$_2$). The average mass loadings of isoprene epoxydiols (~0 ug m$^{-3}$), glyoxal (0-0.1 ug m$^{-3}$), and methylglyoxal (0.1-0.25 ug m$^{-3}$) are comparable to modeled mass loadings of these products in northern Michigan by Li et al. (2015). The composition of semi-volatile SOA is split between hydroperoxides (ROOH), predominately formed by OH oxidation followed by reaction with HO$_2$, and RONO$_2$. Organic nitrates constitute 6% of total SOA above the canopy and 7% below the canopy, in good agreement with results in both the Southeast U.S. (3-8%) (Lee et al., 2016) and a Colorado front-range forest (6-20%) (Fry et al., 2013). Other oxidation products (alcohols, peroxyacetyl nitrates, etc.) contribute a negligible amount of aerosol.

Ultimately, SOA mass loadings above and below the canopy are within 0.5 μg m$^{-3}$ of each other, and this similarity is caused by efficient mixing and different influences on aerosol production partially offsetting each other. For instance, lower temperatures below the canopy cause increased partitioning of semi-volatile oxidation products (ROOH and RONO$_2$) to the particle phase relative to above the canopy, resulting in nighttime mass loadings as much as 35% larger below the canopy than above (when temperature differences are maximized). However, as the majority of semi-volatile products with volatilities low enough to partition appreciably to the aerosol phase contain peroxide functional groups, even in the case of RONO$_2$, lower HO$_x$ concentrations below the canopy partially offset the effect of temperature. This is highlighted by the fact that if temperatures above and below the canopy are assumed to be the same, more SOA mass is produced above the canopy than below (Fig. S11).

In addition, ELVOC products have vapor pressures low enough that virtually all of their mass exists within the particle phase, causing temperature to have little to no effect on their partitioning between gas and aerosol. Therefore, higher rates of $O_3$ oxidation of α-pinene above the canopy further offset the effect of increased partitioning of semi-volatile products below the canopy. Isoprene epoxide, glyoxal, and methylglyoxal mass loadings are only slightly larger below the canopy than above for the majority of the day (within 5-10% of each other) because of lower rates of aerosol deposition and similar gas-phase concentrations in each location (due to mixing), despite the fact that gas-phase epoxide production is more rapid above the canopy.

Overall, $NO_3$-derived products contribute little to total $RONO_2$ SOA mass, causing production of $RONO_2$ SOA to be driven primarily by the OH + NO pathway, even at night (Fig. S12). The small mass loading of $NO_3$-derived products is primarily the result of our assumption that all monoterpenes are α–pinene. As $NO_3$ oxidation of α-pinene produces first-generation nitrates, which have relatively high volatilities, almost exclusively (Berkemeier et al., 2016), the SOA mass yield of $NO_3$ oxidation is relatively small (Table 1). However, previous modeling with a variety of monoterpenes suggests as much as 0.2-0.4 μg m$^{-3}$ of SOA is produced from monoterpene nitrates in northern Michigan (Pye et al., 2015), in agreement with the much higher mass yields of other monoterpenes such as β-pinene and limonene. Adding this mass to the total amount of SOA would suggest $RONO_2$ contributes 16-24% of total SOA mass, on the upper end of estimates from previous measurements (Fry et al., 2013; Lee et al., 2016).

Assuming this amount of monoterpene nitrate SOA mass is present, and assuming that the fraction of $RONO_2$ mass derived from $NO_3$ oxidation is the same as we have modeled for α-pinene (~45-80%) (Fig. S13), $NO_3$ oxidation could be expected to contribute around 4-12% of total SOA mass in this environment. This contribution is reasonable, as Lee et al. (2016) recently found that $RONO_2$ produced from $NO_3$ oxidation contribute around 22-33% of biogenic SOA in a remote coniferous forest in Canada. The sensitivity of $RONO_2$ SOA and SOA from $NO_3$-derived oxidation products to background pollutant levels is shown in Figure S14. Both total $RONO_2$ SOA and $NO_3$-derived SOA increase by a factor of 3-4 under increased pollutant loadings. Interestingly, increases in $NO_3$-derived $RONO_2$ SOA mass are almost solely dependent upon increased $O_3$ mixing ratios, implying that $NO_3$ SOA production is essentially $O_3$-limited for the conditions modeled in the sensitivity analysis.

The preceding analysis serves as an indicator of the potential relevance of $NO_3$ to total aerosol mass in this location; however, our model results also suggest the possibility of net transport of $NO_3$-derived $RONO_2$ from the below-canopy environment through the canopy, potentially resulting in more $NO_3$-derived $RONO_2$ SOA mass than would be assumed if transport were neglected. Indeed, similarly to gas-phase $RONO_2$ concentrations, including in-canopy mixing results in a substantial increase in both isoprene and α-pinene $NO_3$-derived OA above the canopy. This is shown in Figure 12, where positive values indicate increased amounts of SOA mass above the canopy when mixing is included within the model. Ultimately our results suggest a 20-50% increase in $NO_3$-derived α-pinene $RONO_2$ mass in the afternoon, which can be thought of as a surrogate for monoterpene nitrate mass, and an even larger increase

from isoprene products. While we assume no loss of products during transport, which would reduce this effect, these results highlight the potential relevance of the below-canopy environment on above-canopy SOA mass. Because the contribution of $NO_3$-derived $RONO_2$ to total SOA is predicted to be high in areas such as the southeast U.S. (Ayres et al., 2015), the potential influence of net transport of $NO_3$-derived products and the associated effect on SOA mass warrants further study.

## 4. Conclusions

A detailed 0D model was used to investigate α-pinene and isoprene oxidation chemistry and SOA production above and below a forest canopy. Specific focus was placed on the contribution of $NO_3$ to BVOC processes, as shade provided by the canopy was assumed to reduce $NO_3$ photolysis rates. While $NO_3$ mixing ratios are relatively low due to the rural nature of the site (<0.5 pptv), daytime mixing ratios are two-to-three times larger below the canopy than above. Reduced photolysis frequencies are found to be the primary factor behind elevated daytime below-canopy $NO_3$ concentrations. Higher $NO_3$ mixing ratios below the canopy lead to a substantially higher contribution of $NO_3$ to daytime oxidation of α-pinene below the canopy (~12%) than above the canopy (~4%), and this contribution increases to 32% below the canopy under hypothetical conditions meant to represent a polluted suburban environment. While the efficient mixing process parameterized within the model results in similar gas-phase $RONO_2$ concentrations above and below the canopy, our results suggest that daytime net transport of $NO_3$-derived products upward from the below-canopy environment constitutes as much as 30% of total $NO_3$-derived $RONO_2$ production above the canopy in the morning. However, differences between the predictions of gas-phase $RONO_2$ by our MCM-based model, predictions from previous modeling of $RONO_2$ at the same site using laboratory yields (Pratt et al., 2012), and $RONO_2$ measurements (Romer et al., 2016) underscore the need for continued improvement of our understanding of $RONO_2$ chemistry.

Similarly to gas-phase $RONO_2$ mixing ratios, total SOA mass loadings are comparable above and below the canopy. The fact that ELVOCs constitute a majority of SOA mass implies the need for further study of these compounds, as they are not yet included in major chemical mechanisms such as the MCM but are likely highly relevant to SOA formation in rural areas. Lower temperatures below the canopy lead to increased partitioning of oxidation products to the particle phase and therefore higher mass loadings of semi-volatile products; however, the total contribution of these products to SOA mass is relatively low (~10-25%) under the simulated conditions.

Ultimately, substantially more $NO_3$-derived SOA mass from both α-pinene and isoprene oxidation products is modeled above the canopy when transport from the below-canopy environment is included. While our model results predict a small mass loading of $NO_3$-derived products due to the use of α-pinene as the sole monoterpene, Ayres et al. (2015) calculate that 23-44% of $NO_3$ lost to reaction with monoterpenes becomes aerosol phase $RONO_2$, and it has been estimated that $NO_3$-derived $RONO_2$ contribute as much as 19-34% of total OA in the southeast U.S. (Xu et al., 2015). Therefore, our results, when combined with the results of previous studies highlighting the relevance of $NO_3$ on total OA mass

loadings, suggests that regional 3D models that lack the near-ground resolution to account for effects of the forest canopy may be missing a substantial source of OA.

**Copyright Statement:**

Authors grant Copernicus Publications commercial rights to produce hardcopy volumes of the journal for purchase by libraries and individuals.

**Code Availability:**

The MATLAB code associated with this manuscript is available upon request.

**Data Availability:**

The compiled datasets used to produce each figure within this manuscript are available as Igor Pro files upon request.

**Author Contribution:**

B.C. Schulze developed the model, performed simulations and data analysis, and wrote the manuscript. H.W. Wallace and R.J. Griffin assisted heavily with model development, data analysis, and manuscript editing. H.W. Wallace, J. H. Flynn, B.L. Lefer, M.H. Erickson, B.T. Jobson, S. Dusanter, S.M. Griffith, R.F. Hansen, T. VanReken, and P.S. Stevens performed atmospheric measurements during CABINEX that were used as model inputs. J.H. Flynn, P.S. Stevens, S. Dusanter, S.M. Griffith, and R.F. Hansen provided helpful comments and edits.

**Acknowledgements:**

We would like to acknowledge the National Science Foundation for funding this work with grant numbers AGS-0904214 and AGS-0904167.

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

**Table 1.** Overview of SOA mass yields from previous chamber experiments on isoprene, α-pinene, β-pinene, and limonene

| BVOC | SOA mass yield (low-NO$_x$ OH oxidation) | SOA mass yield (NO$_3$ oxidation) |
|---|---|---|
| isoprene | 0.01-0.053[1] | 0.04-0.238[1,2] |
| α-pinene | 0.24-0.36[3] | ~0-0.13[4,8] |
| β-pinene | 0.14-0.17[5] | 0.46-0.53[6] |
| limonene | 0.09-0.34[7] | 0.4[8] |

[1]Kroll et al., 2006
[2]Ng et al., 2008
[3]Eddingsaas et al., 2012
[4]Fry et al. 2014
[5]Sarrafzadeh et al., 2016
[6]Fry et al., 2009
[7]Griffin et al. 1999
[8]Spittler et al. 2006

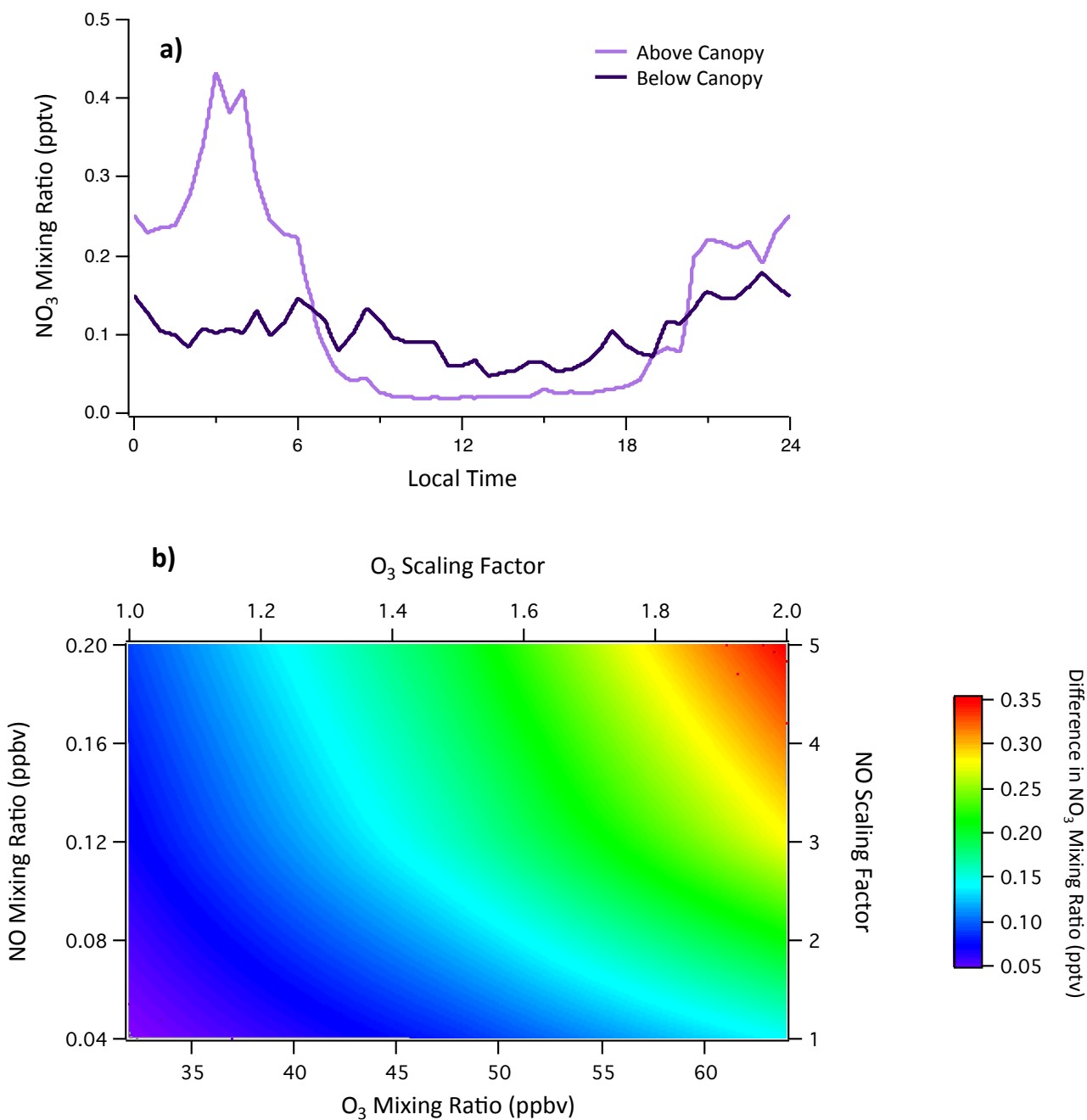

**Figure 1.** (**a**) Modeled $NO_3$ mixing ratios (pptv) above and below the canopy. (**b**) Sensitivity of above-below canopy difference in midday (11:00-15:00) $NO_3$ mixing ratios to changes in $O_3$ and NO concentrations. Positive values indicate larger $NO_3$ mixing ratios below the canopy. The term "scaling factor" specifies changes to the diurnal profiles of $O_3$ and NO, while "mixing ratios" refer to the resulting average mixing ratios of these species between 11:00 and 15:00.

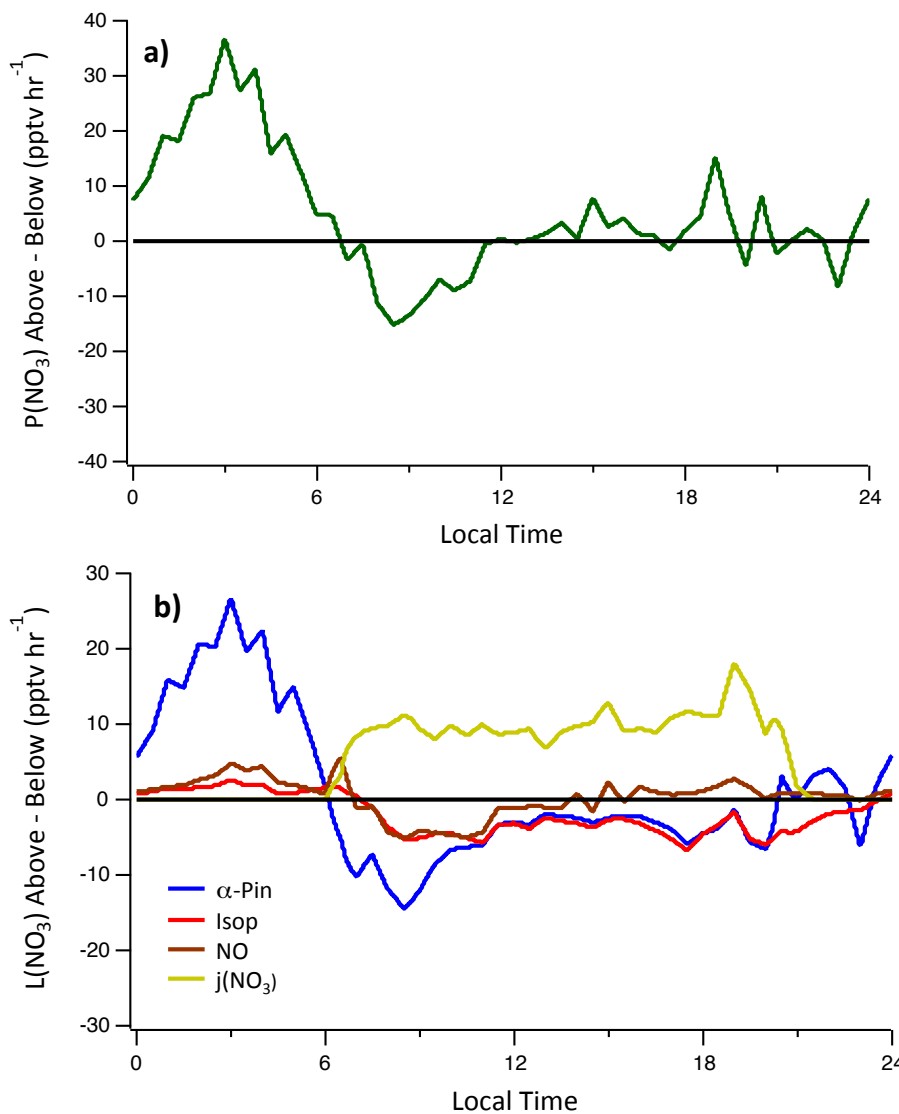

**Figure 2.** (**a**) Difference in NO$_3$ production rate (P(NO$_3$)) above and below the forest canopy. (**b**) Difference in individual NO$_3$ loss rates (L(NO$_3$)) from photolysis, reaction with NO, oxidation of isoprene, and oxidation of α-pinene above and below the forest canopy.

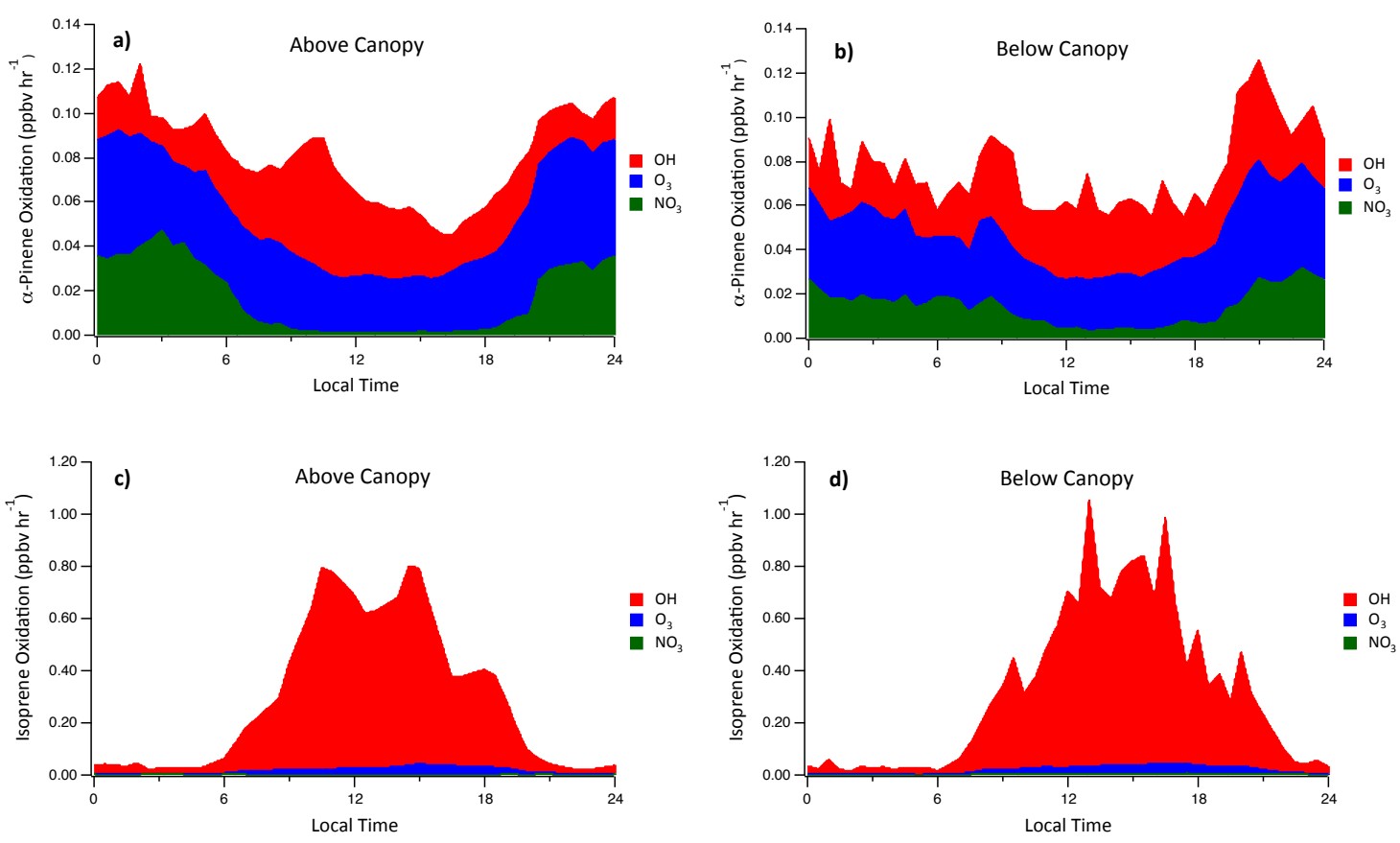

**Figure 3.** Modeled diurnal oxidation rates of (**a**) α-pinene above-canopy, (**b**) α-pinene below-canopy, (**c**) isoprene above-canopy, and (**d**) isoprene below-canopy by OH, O₃, and NO₃.

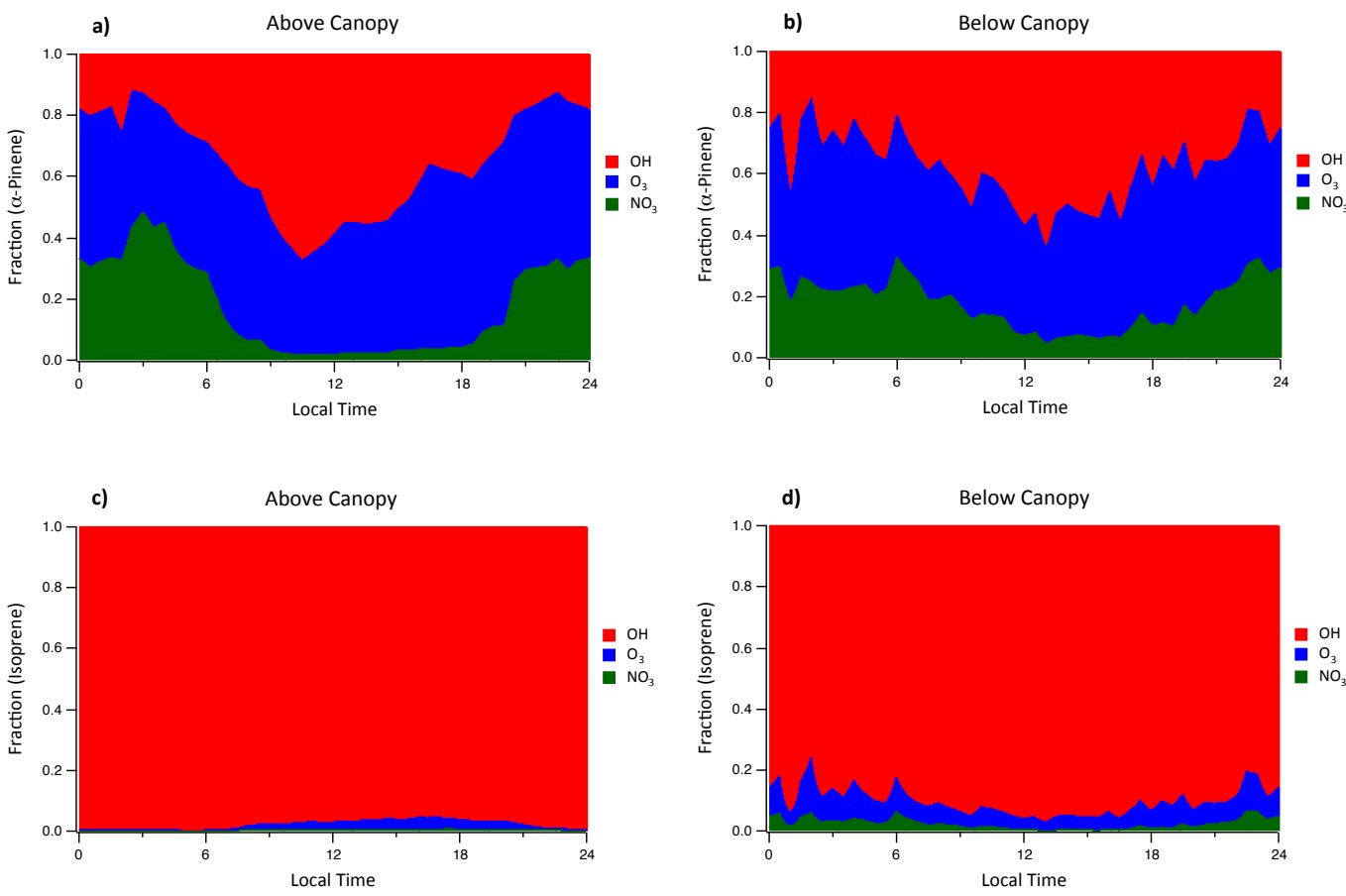

**Figure 4.** Modeled fractional contribution to oxidation rates of (**a**) α-pinene above-canopy, (**b**) α-pinene below-canopy, (**c**) isoprene above-canopy, and (**d**) isoprene below-canopy by OH, O$_3$, and NO$_3$

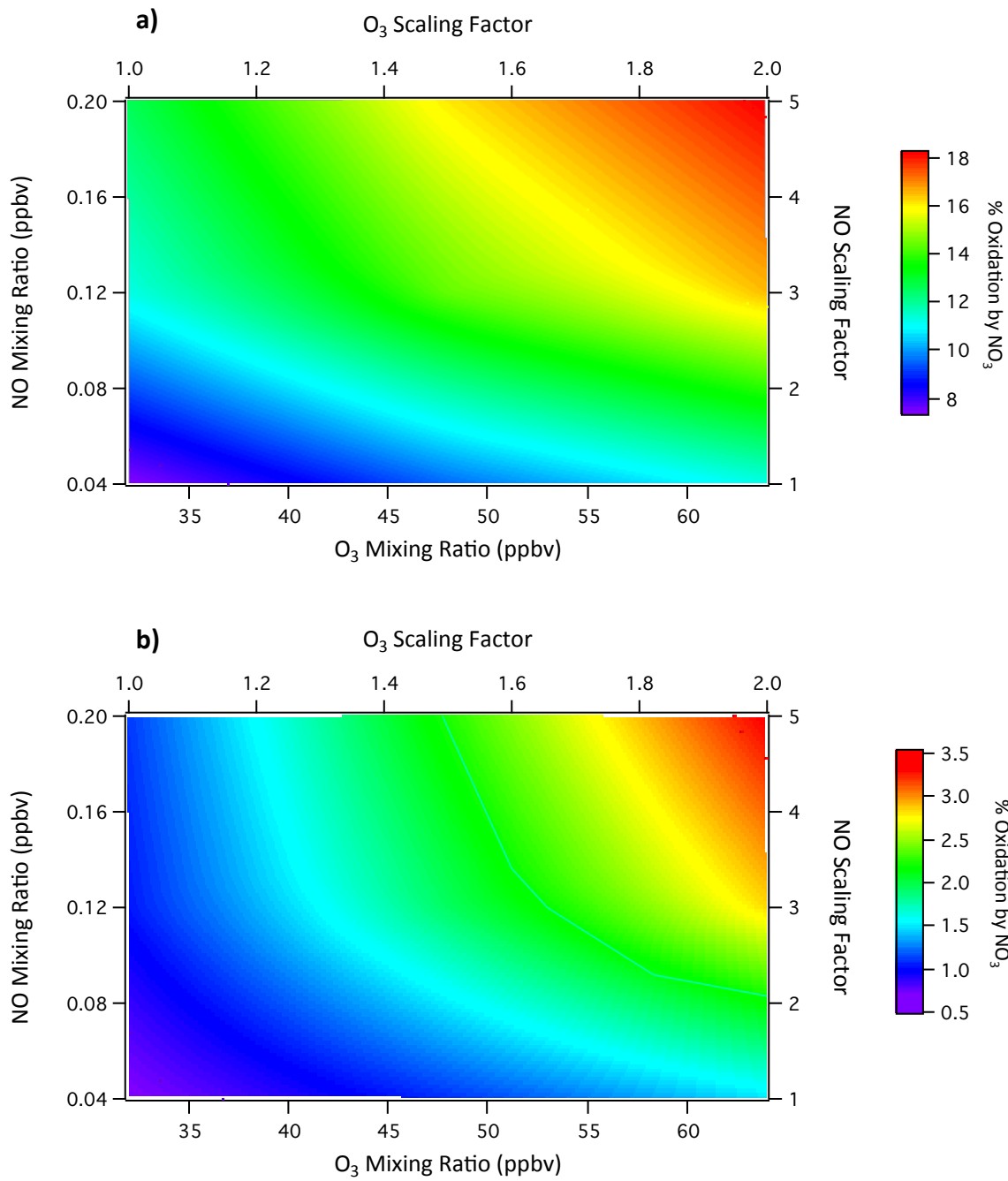

**Figure 5.** Sensitivity of average fractional oxidation of **(a)** α-pinene and **(b)** isoprene by NO₃ below the canopy from 11:00-15:00 to changes in O₃ and NO concentrations. The term "scaling factor" specifies changes to the diurnal profiles of O₃ and NO, while "mixing ratios" refer to the resulting average mixing ratios of these species between 11:00 and 15:00.

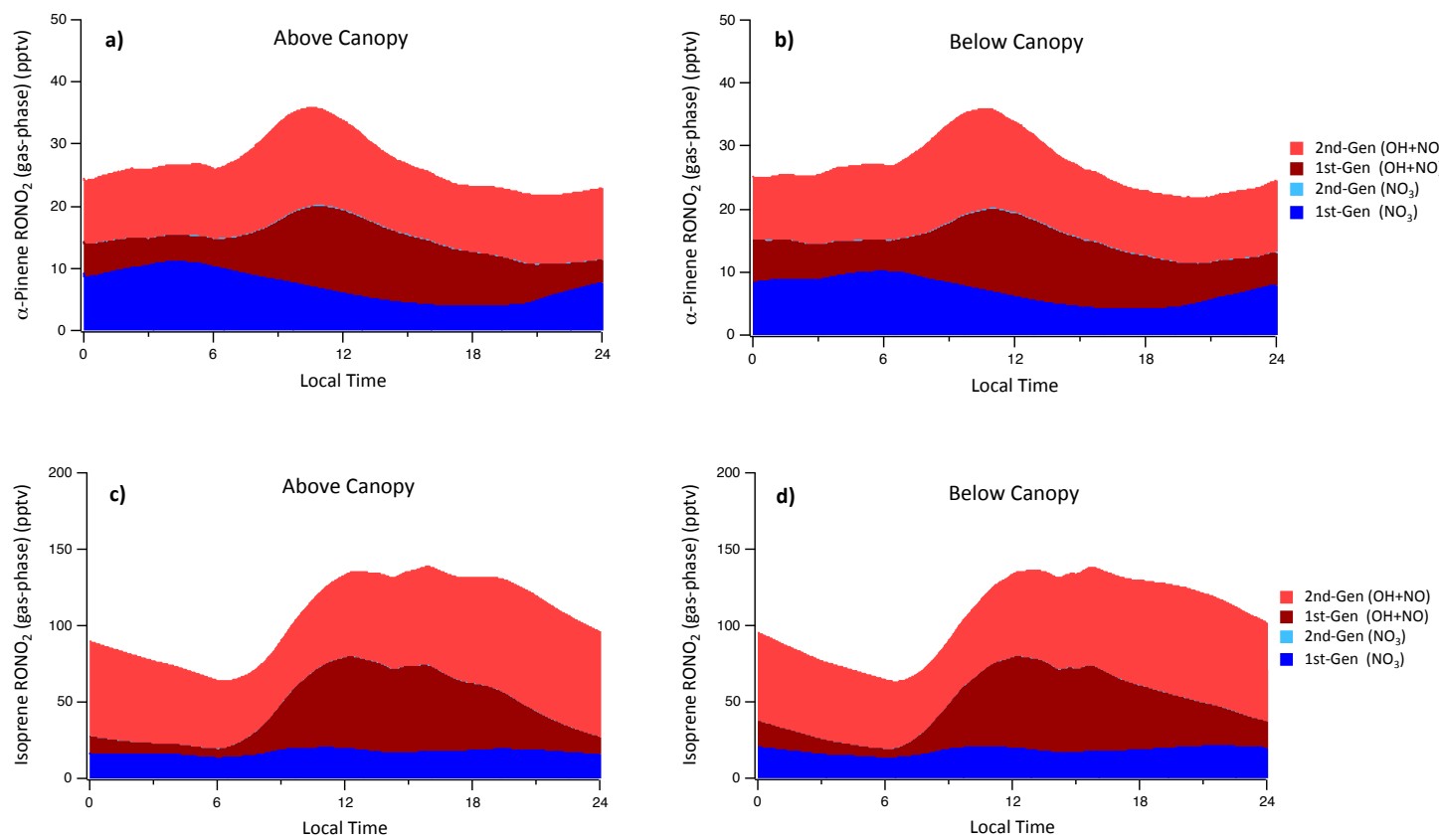

**Figure 6.** Modeled gas-phase RONO$_2$ concentrations from (**a**) α-pinene above-canopy, (**b**) α-pinene below-canopy, (**c**) isoprene above-canopy, and (**d**) isoprene below-canopy. Species produced from NO$_3$ oxidation are blue, while those produced through OH + NO oxidation are red. Darker colors indicate first-generation products; lighter colors indicate second-generation products.

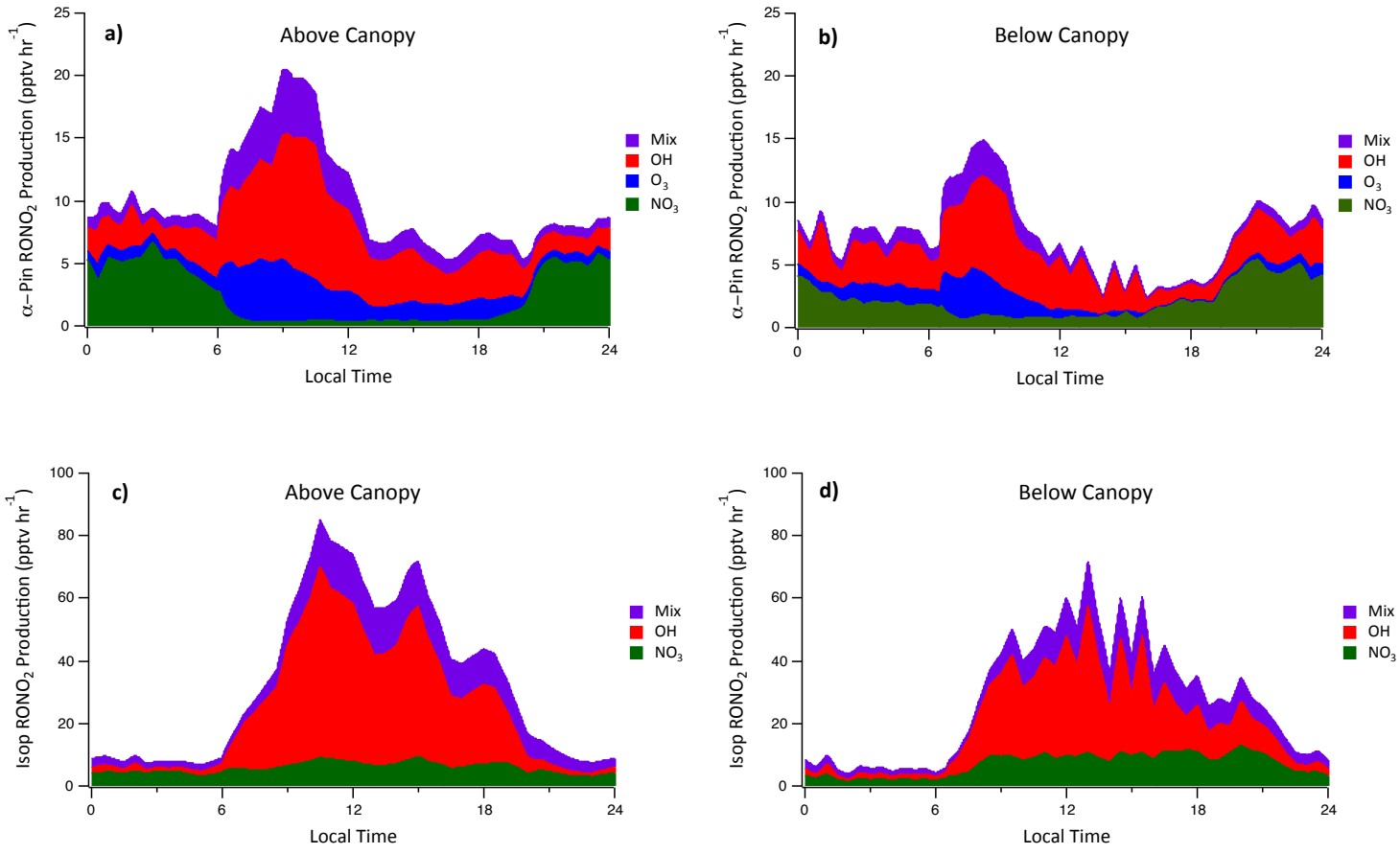

**Figure 7.** RONO$_2$ production through oxidation of (**a**) α-pinene above-canopy, (**b**) α-pinene below-canopy, (**c**) isoprene above-canopy, and (**d**) isoprene below-canopy by NO$_3$, O$_3$, OH and a mix of oxidants. Each color represents the sum of the production rates of each RONO$_2$ species formed through initial oxidation of the parent VOC (isoprene or α-pinene) by the specified oxidant. The "Mix" category represents later-generation RONO$_2$ products formed through initial oxidation by NO$_3$, O$_3$, or OH followed by secondary oxidation by a different oxidant.

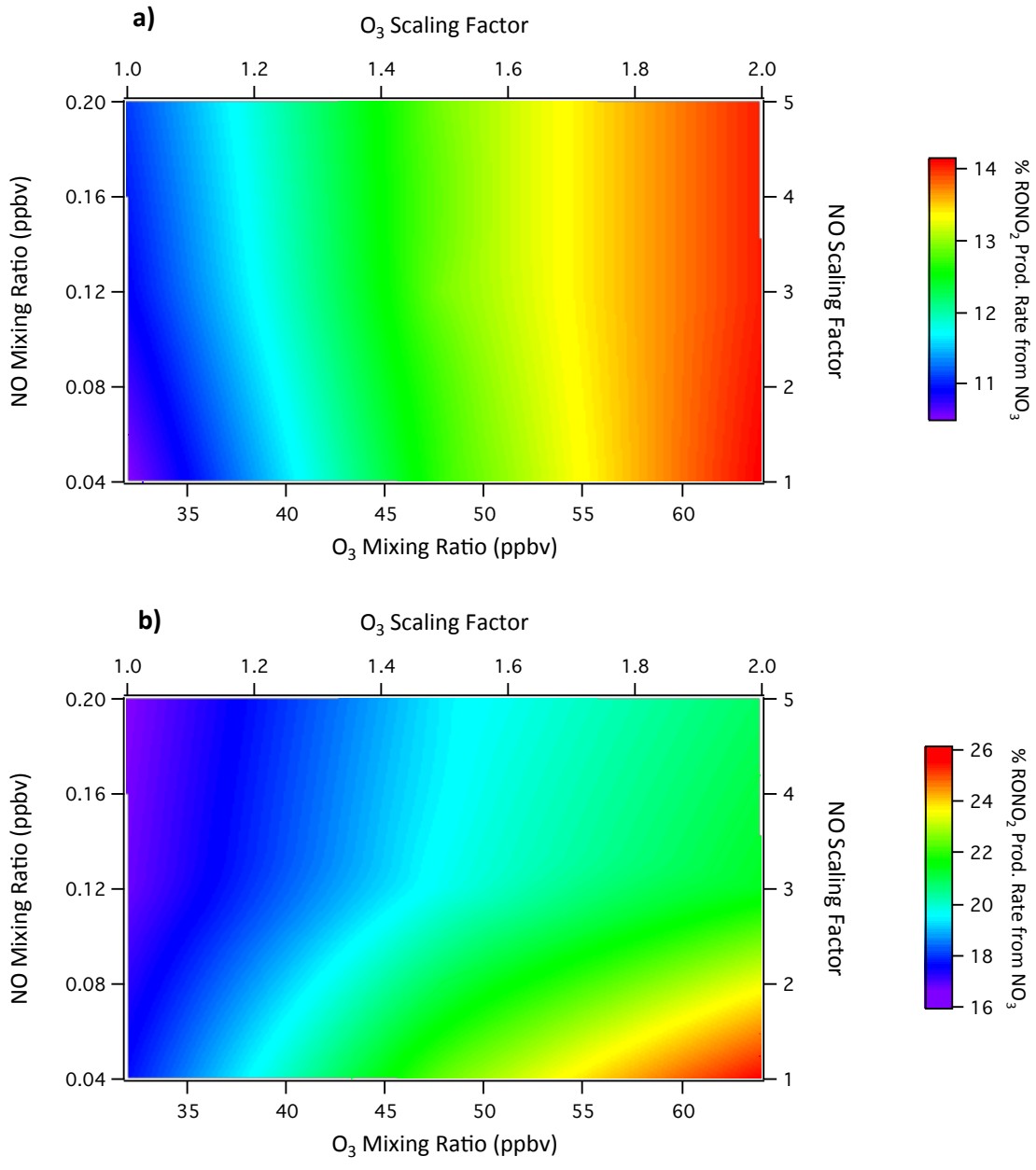

**Figure 8.** Percent of total daytime gas-phase RONO$_2$ production due to NO$_3$ oxidation for varying levels of O$_3$ and NO **(a)** above the canopy and **(b)** below the canopy. Values represent averages from 11:00 to 15:00.

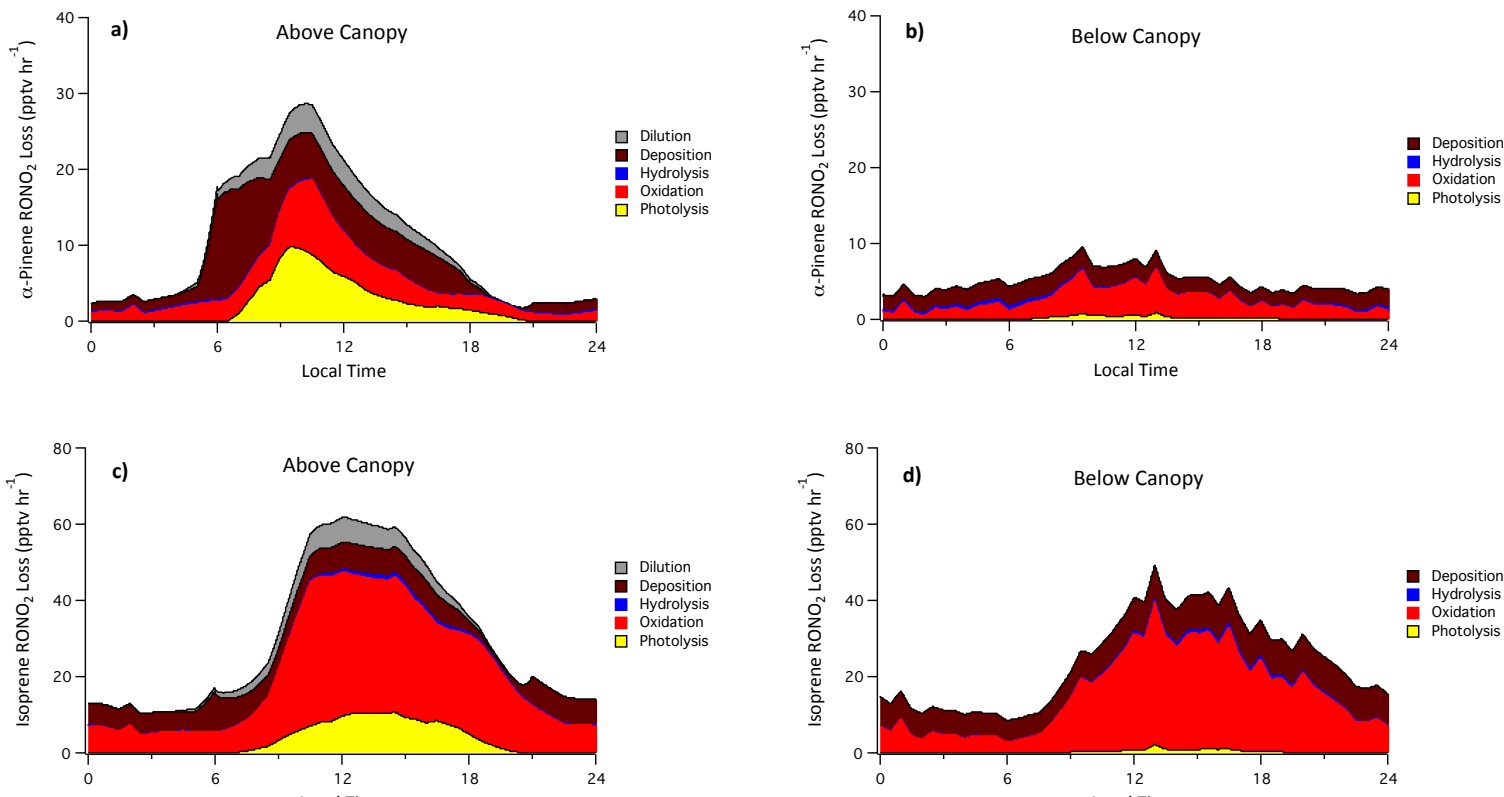

**Figure 9.** Modeled RONO$_2$ loss from photolysis (gas-phase), oxidation (gas-phase), hydrolysis (particle-phase), deposition (gas- and particle-phase), and dilution (gas- and particle-phase) for (**a**) α-pinene products above-canopy, (**b**) α-pinene products below-canopy, (**c**) isoprene products above-canopy, and (**d**) isoprene products below-canopy.

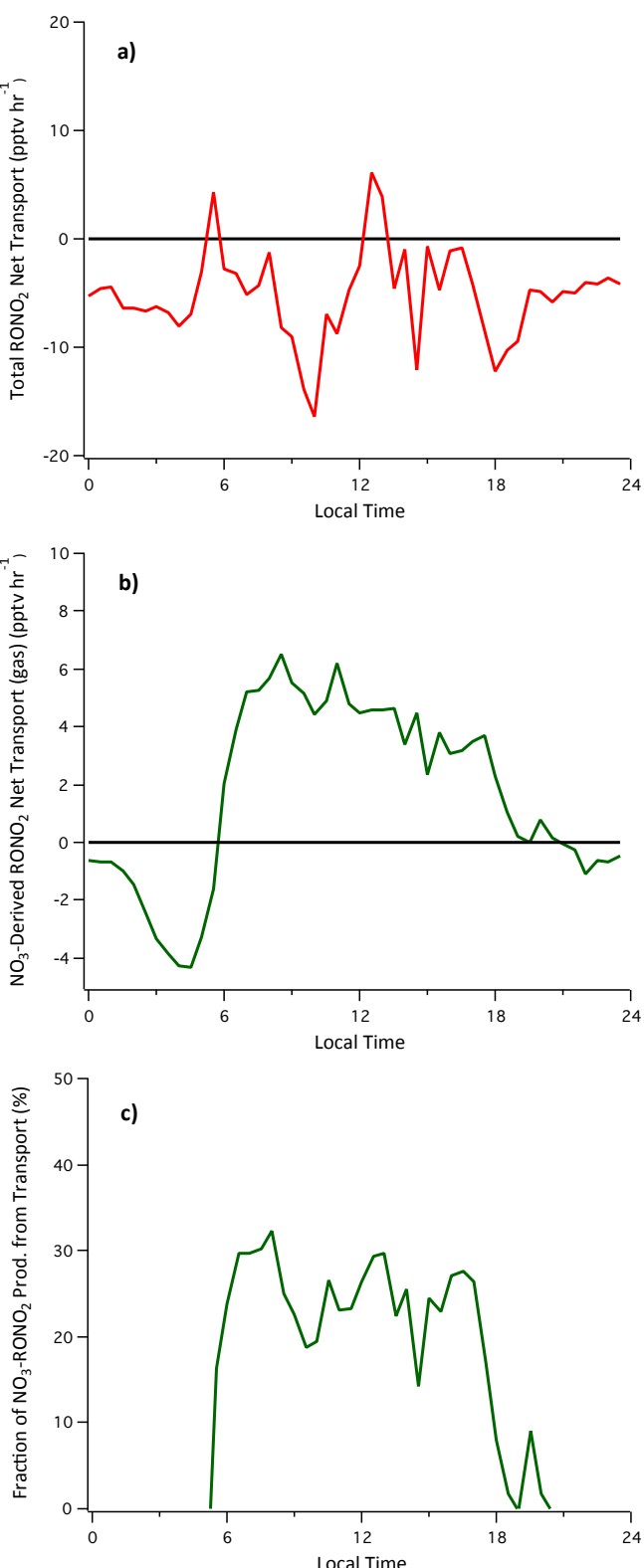

**Figure 10. (a)** Net transport of total gas-phase $RONO_2$ above the canopy. Positive indicates transport from the below-canopy environment; negative indicates transport to the below-canopy environment. **(b)** Same as (a), but for $RONO_2$ produced solely from $NO_3$ oxidation of BVOCs. **(c)** Percent of total production of $NO_3$-derived $RONO_2$ species above the canopy resulting from net transport from the below-canopy environment. This figure implies downward net transport of OH-derived $RONO_2$ species and upward net transport of $NO_3$-derived $RONO_2$ species.

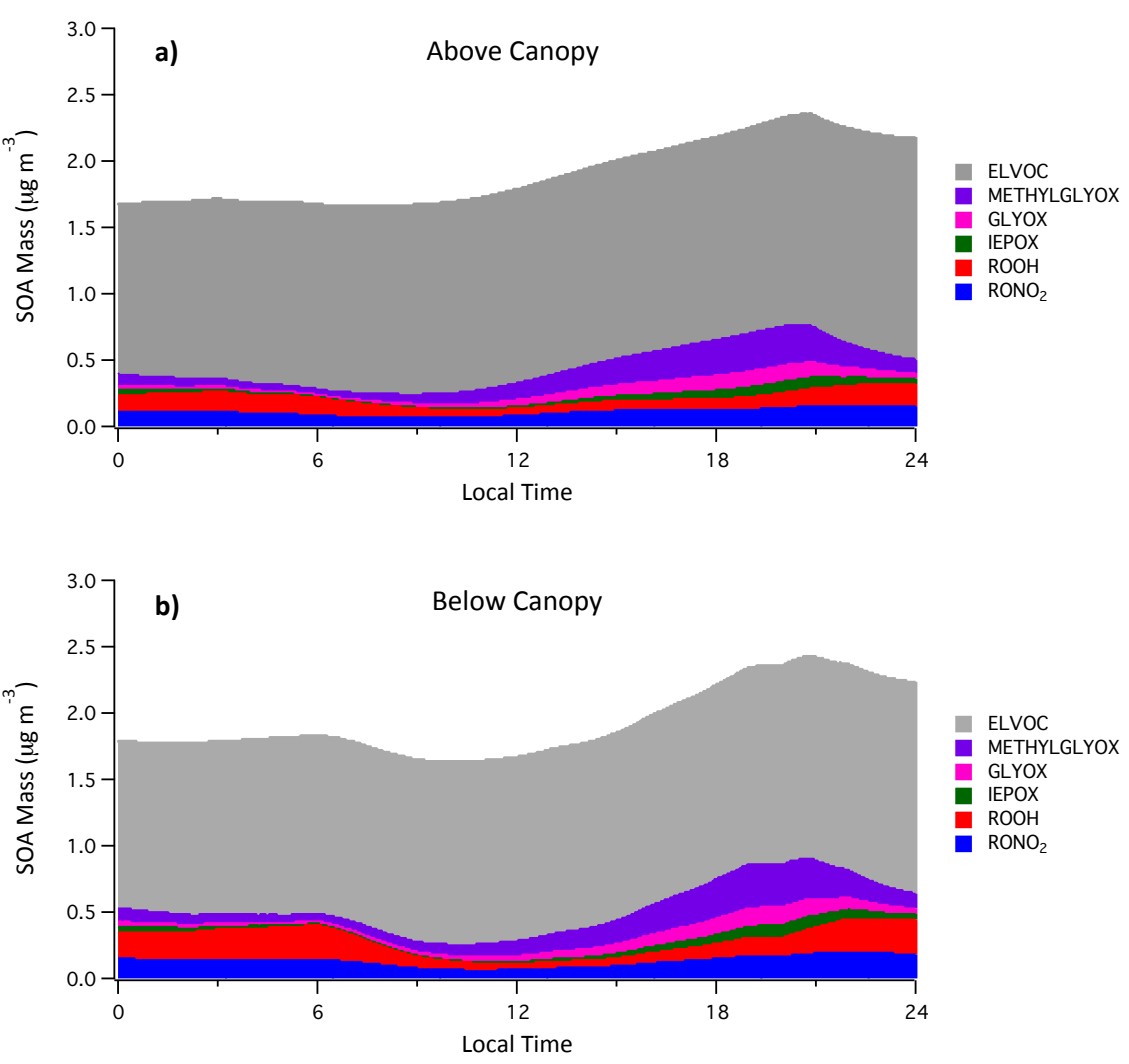

**Figure 11.** Total modeled SOA (**a**) above-canopy and (**b**) below-canopy characterized by fractional contributions from semi-volatile organic nitrates (RONO$_2$) and peroxides (ROOH), isoprene epoxides (IEPOX), glyoxal (GLYOX), methylglyoxal (METHYLGLYOX), and ELVOCs.

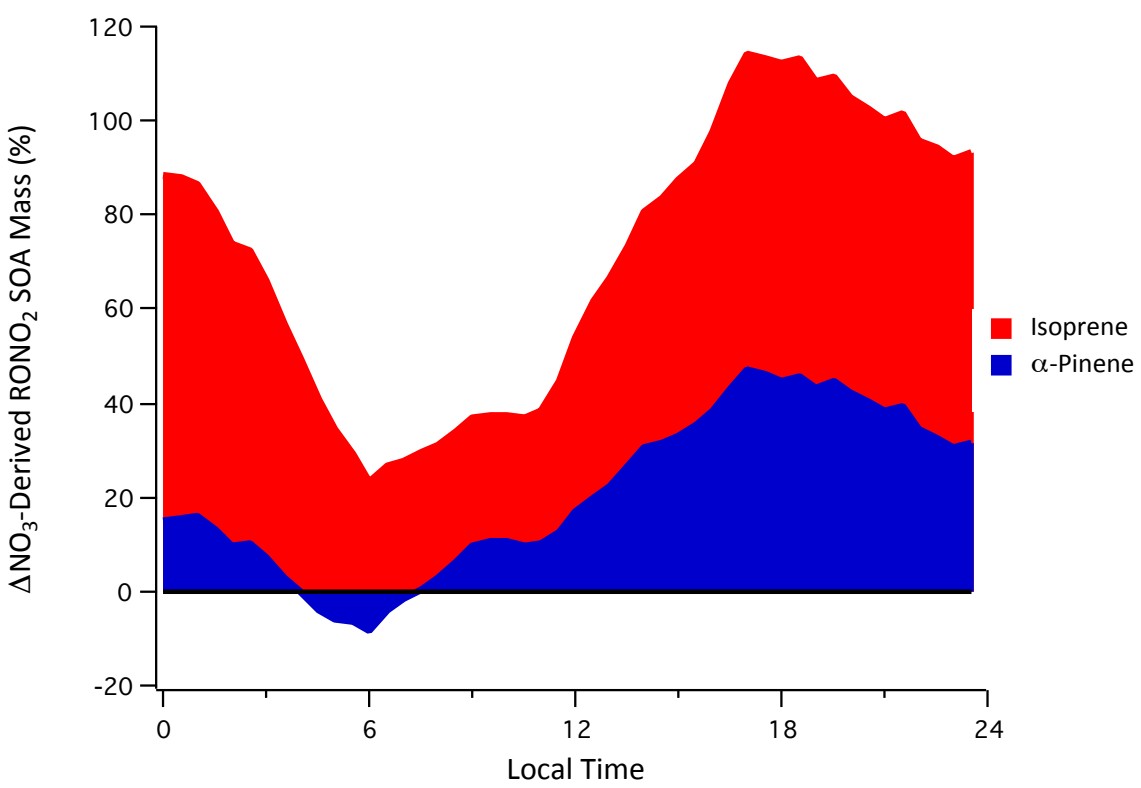

**Figure 12.** Diurnal profile of the net change in NO$_3$-derived RONO$_2$ SOA above the canopy as a result of including transport between the above- and below-canopy environments.