# Peer review of "Differences in BVOC oxidation and SOA formation above"

_Atmospheric Chemistry and Physics, 2016_

## Referee Comment (RC1) · Anonymous Referee #1 · 29 Jul 2016

This paper describes a modeling study of chemistry occurring above and within a forest canopy, with a focus on understanding differences that arise between the canopy and above-canopy regions as well as between different NOx levels. The authors construct two 0-D box models using the Master Chemical Mechanism (MCM) to describe the oxidation of isoprene and monoterpenes by OH, O3, and NO3 over the course of an entire day. The authors emphasize the importance of NO3 chemistry, pathways that form organic nitrates, and use an equilibrium partitioning model to predict SOA formation from the oxidation products.

Overall this study has some nice insights and could be suitable for publication in ACP, but only after some fairly major concerns are addressed. These concerns are listed in no particular order.

[Figure]

1. The validity of 2 0D box models. Surely the authors are aware that there is efficient mixing of below canopy air with above canopy air, and vice versa. This fact is apparently not mentioned (or at least not thoroughly discussed), and without some additional discussion and figures of actual data, the modeling results are very difficult to interpret. The canopy layer makes up <10% of the full boundary layer, so are findings of production rates, etc normalized or corrected for this small contribution to the mean boundary layer mass concentration? Presumably the observations used to constrain the model are vertically resolved but no vertical profiles of the observations are shown to evaluate how representative the two boxes used for modeling actually are. Oxidation products formed below the canopy, or above, would cycle many times between the two regions over the course of a day, scrambling the signatures and likely diluting the effect of chemistry occurring within the canopy itself.

2. While the MCM is becoming more useful, it is still rather a dangerous model to use for SOA studies because the mechanisms are not complete at the 5% level of carbon mass, but that 5% might be 90% of carbon capable of forming SOA. For example, the MCM does not include auto-oxidation of monoterpenes (Ehn et al Nature 2014 and others) - thereby missing a likely significant fraction of SOA mass, as well as highly oxygenated nitrates (Lee et al PNAS 2016). How might this affect the conclusions?

3. Lack of deposition. Deposition of oxidation products appears not to be included. This seems rather problematic for interpreting the SOA formation potential. Deposition in the canopy will be much more significant than above the canopy, at least during nighttime, and during daytime deposition from both would significantly impact the available vapors for SOA formation, especially given the equilibrium partitioning assumption used for modeling SOA. With any horizontal wind through the canopy, there will presumably be a significant depositional sink given the proximity to canopy elements. What is the lifetime of a product formed in the canopy against deposition compared to mixing out of the canopy layer or to SOA formation? How was the vertical distribution of the condensation sink constrained?

4. Given 2 and 3 above, the discussion on SOA seems rather limited in its usefulness to actual SOA formation potential. In addition, more interesting would be to incorporate the lapse rate and vertical mixing impacts on SOA formation given the cycling of oxidation products across a 10K temperature gradient.

5. If I understand correctly, RONO2 form $\sim$45% of the SOA mass in the model. In the SE U.S., it was more like 3-10% (Xu et al PNAS 2015, Lee at al PNAS 2016). The authors cite Fry et al, and note consistency with that study, but, are the distributions of BVOC emissions at all similar between Manitou and UMBS? Presumably the high fraction of particulate RONO2 at Manitou was the dominance of monoterpenes, is that the case at UMBS?

6. Odd model set up choices. The "urban" case, which apparently mixes data from Detroit and Houston is a bit odd. Why not make it purely hypothetical? I don't see why the authors fix the NO/NO2 ratio - that would seem to be a good metric to test the chemistry in the model as it is sensitive to total RO2, not just HO2 and O3. Moreover, as NOx increases the NO/NO2 ratio isn't going to be fixed, there are important feedbacks between O3, HO2, RO2, and NO concentrations that are important and why one would want to use a model in the first place. Fixing the NO/NO2 ratio makes nitrate formation rates a linear extrapolation with increasing NOx when the non-linear couplings of HOx and NOx might lead to a different result than found here.

---

## Referee Comment (RC2) · Anonymous Referee #2 · 29 Jul 2016

General Description of the manuscript:

The authors use above- and below-canopy measurements of atmospheric composition and a 0D box model to evaluate differences in the oxidative fate and secondary organic aerosol (SOA) formation of the biogenic volatile organic compounds isoprene and alpha-pinene. The authors find that nitrate (NO3) concentrations are higher below the canopy than above, leading to enhancements in first-generation organic nitrate (RONO2) concentrations. SOA mass concentrations are also higher below the canopy than above, as the ratio of hydroperoxyl radical (HO2) to nitrogen oxide (NO) is higher below the canopy than above.

The article is suitable for publication in ACP, but after major revision.

General Comments:

[Figure]

–> The authors dedicate lengthy discussions to obvious and/or unsurprising results. Some examples include: (1) that isoprene oxidation is predominantly by reaction with OH is mentioned in the abstract and has a dedicated paragraph (p. 11, lines 23-29) and figure (Figure 6); (2) an entire section is dedicated to detailing the enhancement in daytime nitrate concentrations (Section 3.2, p. 10), but this can be reduced to 1-2 sentences pointing to the importance of photolysis; and (3) page 8, lines 30-34: that NO3 and O3 perturbations change NO3 is obvious and indicated in Equation (4).

–> Steps in the methodology are not well justified and so appear arbitrary. For example, the authors include observations from a site in Detroit and also appear to impose an imaginary forest canopy in the 0D simulations. The analysis of the CABINEX site under polluted conditions seems sufficient to show the effect of pollution on RONO2 and SOA formation above and below a forest canopy.

–> It is not clear from the abstract or concluding statements what the impact is of the results from this study. Are there any implications in this study for greening urban areas or for rapid urbanization (that is, rapid land cover change in and around urban areas)? In the Short Summary the authors mention that it is important to understand the impact of forest canopies on the oxidative fate and SOA formation of reactive VOCs "as forested areas downwind of urban areas (and therefore the residents) will be impacted by this phenomenon." How will they be impacted?

Specific Comments:

–> Page 2, line 2: The start of the sentence "The most significant first-generation RONO2 formation mechanism" is ambiguous. Do the authors mean that most RONO2 is first-generation RONO2 or is this referring to a specific first-generation RONO2 species that dominates?

–> Page 2, lines 28-29: Will the 0D model in this study then also underestimate OA mass loadings?

–> Page 3, lines 20-21: Are the authors referring to a previous study when they state: "It has been further hypothesized that"? If so, then please provide the reference.

–> Page 3, line 37: Please briefly indicate for the reader what the differences are in BVOC (isoprene and alpha-pinene) reactivity and SOA formation.

–> Page 4, lines 21-24: Please indicate what the effect is of not accounting for liquid water content and particle-phase reactions on SOA predicted by the model.

–> Page 6, line 1: "...varies over the range of 0 to 0.17." This range includes night time, but more useful for the reader is the variability in this ratio during daylight hours when photolysis is occurring.

–> Page 6, lines 1-2: "Model input data are further described in the SI". There is no description of the model input data in the SI; only figures and captions.

–> Page 6, line 13: "In the second modeled case...". What was the first modeled case? This is not systematically presented in the methods section.

–> Page 6, lines 34-35: Please indicate in the text why Houston sites were selected to obtain NO2/NO ratios for Detroit. Are NO2/NO ratios reasonably similar for all urban sites?

–> Page 7, line 4: "The Detroit data display...". Please indicate in the text where this data is displayed.

–> Page 7, line 12: Please point the reader to Figure S4 at the start of paragraph, rather than midway through. This provides context for the discussion.

–> Page 7, lines 14-15: "The model tends to under predict nighttime OH concentrations". Is this after taking into consideration measurement interference?

–> Page 8, lines 4-5: It's not apparent how the results support an isoprene-derived measurement interference. The slope is near unity (Figure S5(b)) when the model does not include interfering isoprene RO2, but less than unity (slope = 0.7) when it

does. This would suggest that the interference is negligible.

–> Page 12, lines 29-32: More appropriate to compare your total OA (and not SOA only) with total OA from Delia (2004), as the contribution of SOA to total OA is known (page 12, line 37 and page 13, line 1).

–> Page 13, lines 8-9: Does the difference in photolysis above and below the canopy impact SOA formation? It is not apparent that this has been tested in the sensitivity simulations (Figure S6).

–> The authors provide labels for above canopy and below canopy data in panel "(a)" of Figures 2 and S2, that seems to suggest the dark lines are for below canopy data and the lighter lines for above canopy data. If this is the case, the authors should clarify this convention in the figure captions.

---

## Short Comment (SC1) · 9 Aug 2016

I would like to draw the authors' attention to a paper where a similar analysis was made on the importance of daytime $NO_3$ chemistry within a forest (Solar spectral actinic flux and photolysis frequency measurements in a deciduous forest, J. Geophys. Res., 111, D15303, doi:10.1029/2005JD006902, 2006). The conclusion was that because of transport related short residence times of biogenic VOCs, daytime in-canopy degradation was of minor importance.

---

## Author Comment (AC2) · 30 Sep 2016

We thank both reviewers for their insightful comments regarding the manuscript. Many of the issues addressed had a substantial impact on the modeling framework and subsequently the results. These are described in detail below.

REVIEWER #2

1. Referee:

The authors dedicate lengthy discussions to obvious and/or unsurprising results. Some examples include: (1) that isoprene oxidation is predominantly by reaction with OH is mentioned in the abstract and has a dedicated paragraph (p. 11, lines 23-29) and figure (Figure 6); (2) an entire section is dedicated to detailing the enhancement in daytime

nitrate concentrations (Section 3.2, p. 10), but this can be reduced to 1-2 sentences pointing to the importance of photolysis; and (3) page 8, lines 30-34: that NO3 and O3 perturbations change NO3 is obvious and indicated in Equation (4).

Author Response:

We agree that shortening or removing these specific cases (which were included for completeness and proof of model performance) is an appropriate change. With regard to the section describing NO3 concentrations, because NO3 is sensitive to both photolysis rates and O3 and NOx profiles, which are substantially different above and below the canopy, we considered it appropriate to include a short analysis highlighting the specific times when reductions in photolysis caused the elevation in NO3 concentrations, rather than simply claim that this was the case during the middle of the day.

Resulting changes in manuscript:

The revised manuscript will show that: - The mention of isoprene oxidation has been removed from the abstract, as has most of the original paragraph describing isoprene oxidation. Only one sentence has been included at the end of the previous paragraph. - The section describing the daytime enhancement in NO3 concentrations below the canopy has been reduced and added to the section regarding NO3 concentrations. - The section describing the NO3 steady state analysis has been removed.

2. Referee:

Steps in the methodology are not well justified and so appear arbitrary. For example, the authors include observations from a site in Detroit and also appear to impose an imaginary forest canopy in the 0D simulations. The analysis of the CABINEX site under polluted conditions seems sufficient to show the effect of pollution on RONO2 and SOA formation above and below a forest canopy.

Author Response:

Please see response to Reviewer #1, Point #6.

3. Referee:

It is not clear from the abstract or concluding statements what the impact is of the results from this study. Are there any implications in this study for greening urban areas or for rapid urbanization (that is, rapid land cover change in and around urban areas)? In the Short Summary the authors mention that it is important to understand the impact of forest canopies on the oxidative fate and SOA formation of reactive VOCs "as forested areas downwind of urban areas (and therefore the residents) will be impacted by this phenomenon." How will they be impacted?

Author Response:

We acknowledge that a discussion of impacts of the study could be expanded. Ultimately, we conclude that the elevation of NO3 below the canopy influences organic nitrate production rates substantially during the day, leading to a different suite of organic nitrate products above and below the canopy. The effect of elevated NO3 concentrations on SOA formation becomes more significant under polluted conditions with a favorable NO2/NO ratio, which could indicate that forested areas on the outskirts of urban environments may experience substantially higher SOA mass loadings below the canopy than above the canopy, potentially affecting the health of local communities.

Resulting changes in manuscript:

Both the abstract and the conclusions in the updated manuscript will make the implications of the results more explicit.

SPECIFIC COMMENTS, REVIEWER #2

4. Referee:

Page 2, line 2: The start of the sentence "The most significant first-generation RONO2 formation mechanism" is ambiguous. Do the authors mean that most RONO2 is first-generation RONO2 or is this referring to a specific first-generation RONO2 species that dominates?

Author Response/Changes in Manuscript:

Because of the large number of changes incorporated into the model based on both reviewers' suggestions, especially with regard to the sensitivity analysis, this sentence will no longer be relevant.

5. Referee:

Page 2, lines 28-29: Will the 0D model in this study then also underestimate OA mass loadings?

Author Response/Changes in Manuscript:

Our model may underestimate OA mass loadings, as understanding of oxidation, partitioning, and particle-phase chemical mechanisms (oligomerization, the effect of liquid water) are not yet complete enough to ensure entirely accurate results. The aim here is to highlight the potential importance of NO3 chemistry. However, the inclusion of recently discovered pathways for SOA formation (ELVOC formation and reactive uptake of isoprene products) reduces the amount of under-prediction. Furthermore, by ensuring that we under-predict rather than over-predict SOA mass loadings, we can be confident that our conclusions are conservative with respect to SOA formation. These points will be emphasized in the description of limitations of the SOA modeling in the updated manuscript.

6. Referee:

Page 3, lines 20-21: Are the authors referring to a previous study when they state: "It has been further hypothesized that"? If so, then please provide the reference.

Author Response/Changes in Manuscript:

The mentioned sentence has been altered to read, "Furthermore, the majority of these studies have modeled relatively remote locations, encouraging an evaluation of the sensitivity of modeled results to background pollutant concentrations."

7. Referee:

Page 3, line 37: Please briefly indicate for the reader what the differences are in BVOC (isoprene and alpha-pinene) reactivity and SOA formation.

Author Response/Changes in Manuscript:

The following sentences have been added to the manuscript at the appropriate location:

"For instance, isoprene is known to react predominately with OH, while $\alpha$-pinene reacts substantially with all three major oxidants (Fuentes et al., 2006). The SOA yields measured during the photooxidation of isoprene are generally lower than those from $\alpha$-pinene (0-0.053 for isoprene; 0.06-0.21 for $\alpha$-pinene); however, their SOA yields are similar under conditions of NO3 oxidation (0.04-0.238 for isoprene and 0.04-0.16 for $\alpha$-pinene) (Spittler et al., 2006; Ng et al., 2007; Carlton et al., 2009). As $\alpha$-pinene is much more likely to react with NO3 than isoprene, this study highlights potential differences that increased concentrations of NO3 can have on SOA formation from different VOCs."

8. Referee:

Page 4, lines 21-24: Please indicate what the effect is of not accounting for liquid water content and particle-phase reactions on SOA predicted by the model.

Author Response/Changes in Manuscript:

The following sentences have been added to the updated manuscript.

"As these particle-phase reactions can rapidly produce high MW, low volatility compounds, often more aerosol mass is produced in the environment than equilibrium partitioning of the gas-phase species would predict (Johnson et al., 2005; Kroll and Seinfeld, 2008). Furthermore, as increases in RH, and subsequent increases in the water content of SOA, are known to enhance the partitioning of organic species to the aerosol phase, the omission of this process also potentially leads to underestimation

of the total amount of SOA formed (Hennigan et al., 2008). However, as aerosol liquid water is generally driven by inorganic aerosol components (Carlton and Turpin, 2013), which only comprised a small fraction of total aerosol mass during CABINEX, the overall effect of RH is predicted to be small for the conditions of this study (Malm and Day, 2001; Hennigan et al., 2008; VanReken et al., 2015). Therefore, the primary uncertainty in aerosol formation is related to the production of high MW compounds through particle-phase reactions, but these compounds are partially accounted for through the parameterization of ELVOC formation."

9. Referee:

Page 6, line 1: "...varies over the range of 0 to 0.17." This range includes night time, but more useful for the reader is the variability in this ratio during daylight hours when photolysis is occurring.

Author Response/Changes in Manuscript:

The above has been changed to "...varies over the range of 0.05 to 0.17 during daylight hours."

10. Referee:

Page 6, lines 1-2: "Model input data are further described in the SI". There is no description of the model input data in the SI; only figures and captions.

Author Response/Changes in Manuscript:

The above has been changed to, "Model input data are shown in the figures in the SI."

11. Referee:

Page 6, line 13: "In the second modeled case...". What was the first modeled case? This is not systematically presented in the methods section.

Author Response/Changes in Manuscript:

As our method of testing the sensitivity of the results has changed from investigating specific scenarios to evaluating the results more generally using a sensitivity analysis, this sentence no longer applies in the updated manuscript.

12. Referee:

Page 6, lines 34-35: Please indicate in the text why Houston sites were selected to obtain NO2/NO ratios for Detroit. Are NO2/NO ratios reasonably similar for all urban sites?

Author Response/Changes in Manuscript:

As with the last comment, this comment no longer applies in the updated setup of the study.

13. Referee:

Page 7, line 4: "The Detroit data display...". Please indicate in the text where this data is displayed.

Author Response/Changes in Manuscript:

As with the last comment, this comment no longer applies in the updated setup of the study.

14. Referee:

Page 7, line 12: Please point the reader to Figure S4 at the start of paragraph, rather than midway through. This provides context for the discussion.

Author Response/Changes in Manuscript:

The reference to the Figure S4 has been added to the end of the first sentence in the paragraph.

15. Referee:

Page 7, lines 14-15: "The model tends to under predict nighttime OH concentrations". Is this after taking into consideration measurement interference?

Author Response/Changes in Manuscript:

The interference involved in the IU-FAGE measurement affected only measured $HO_2$ concentrations during CABINEX (Griffith et al., 2013). This point is clarified within the section in the updated manuscript.

16. Referee:

Page 8, lines 4-5: It's not apparent how the results support an isoprene-derived measurement interference. The slope is near unity (Figure S5(b)) when the model does not include interfering isoprene $RO_2$, but less than unity (slope = 0.7) when it does. This would suggest that the interference is negligible.

Author Response/Changes in Manuscript:

In this case, because the isoprene-derived interference has been well-characterized elsewhere (Fuchs et al., 2011; Griffith et al., 2013), we assume that the agreement seen between our model and the measurements in terms of $HO_2$ concentrations is due to an over-prediction of $HO_2$ by our model. The fact that the diurnal profiles are so similar (when not including isoprene $RO_2$) is therefore likely somewhat of a coincidence. We will clarify this fact within the updated manuscript; however, the agreement between our model and measured concentrations of $HO_x$ is still well within the range of previously published models (Lelieveld et al., 2008; Pugh et al., 2010; Stavrakou et al., 2010).

17. Referee:

Page 12, lines 29-32: More appropriate to compare your total OA (and not SOA only) with total OA from Delia (2004), as the contribution of SOA to total OA is known (page 12, line 37 and page 13, line 1).

Author Response/Changes in Manuscript:

We have altered the sentence to compare our SOA+POA with the OA measurements made by Delia (2004).

18. Referee:

Page 13, lines 8-9: Does the difference in photolysis above and below the canopy impact SOA formation? It is not apparent that this has been tested in the sensitivity simulations (Figure S6).

Author Response/Changes in Manuscript:

It is clear that the photolysis of NO3 impacts SOA above and below the canopy. However, it also is likely that photolysis of oxidation products themselves affects SOA formation. Based on the assumption that the above-canopy radiation is the upper bound, we have included an investigation of how reducing photolysis rates affects SOA in the sensitivity analysis section of the SOA results.

19. Referee:

The authors provide labels for above canopy and below canopy data in panel "(a)" of Figures 2 and S2 that seems to suggest the dark lines are for below canopy data and the lighter lines for above canopy data. If this is the case, the authors should clarify this convention in the figure captions.

Author Response/Changes in Manuscript:

The reviewer's interpretation of the lines is correct. We have edited the figure captions to clarify the coloring conventions.

References Cited (for replies to both reviewers)

Ashworth, K., Chung, S. H., McKinnet, K. A., Liu, Y., Munger, B. J., Martin, S. T., and Steiner, A. L.: Modelling bi-directional fluxes of methanol and acetaldehyde with the

FORCAsT canopy exchange model, Atmos. Chem. Phys. Discuss., doi:10.5194/acp-2016-522, in review, 2016.

Bean, J. and Hildebrandt Ruiz, L.: Gas-particle partitioning and hydrolysis of organic nitrates formed from the oxidation of $\alpha$-pinene in environmental chamber experiments, Atmos. Chem. Phys., 16, 2175-2184, doi:10.5194/acp-16-2175-2016, 2016.

Bohn, B.: Solar spectral actinic flux and photolysis frequency measurements in a deciduous forest, J. Geophys. Res., 111, D15303, doi:10.1029/2005JD006902, 2006.

[revised manuscript text omitted]

Strong, C., Fuentes, J.D., and Baldocchi, D.: Reactive hydrocarbon flux footprints dur-

ing canopy senescence. Agric. For. Meteorol. 127, 159–173, 2004

VanReken, T. M., Mwaniki, G. R., Wallace, H. W., Pressley, S. N., Erickson, M. H., Jobson, B. T., and Lamb, B. K.: Influence of air mass origin on aerosol properties at a remote Michigan forest site, Atmos. Environ., 107, 35–43, doi:10.1016/j.atmosenv.2015.02.027, 2015.

Wolfe, G. M., Thornton, J. A., Bouvier-Brown, N. C., Goldstein, A. H., Park, J.-H., McKay, M., Matross, D. M., Mao, J., Brune, W. H., LaFranchi, B. W., Browne, E. C., Min, K.-E., Wooldridge, P. J., Cohen, R.C., Crounse, J. D., Faloona, I. C., Gilman, J. B., Kuster, W. C., de Gouw, J. A., Huisman, A., and Keutsch, F. N.: The Chemistry of Atmosphere-Forest Exchange (CAFE) Model – Part 2: Application to BEARPEX-2007 observations, Atmos. Chem. Phys., 11, 1269-1294, doi:10.5194/acp-11-1269-2011, 2011.

---

## Author Comment (AC3) · 30 Sep 2016

Thank you for pointing this out. This important manuscript is now cited, as referenced in our response to reviewer #1.

---

## Author Response (AR1)

Author Response to Reviews – Schulze et al., Differences in BVOC oxidation and SOA formation above and below the forest canopy, ACPD

We thank both reviewers for their insightful comments regarding the manuscript. Many of the issues addressed had a substantial impact on the modeling framework and subsequently the results. These are described in detail below.

REVIEWER #1

1. Referee:

The validity of 2 0D box models. Surely the authors are aware that there is efficient mixing of below canopy air with above canopy air, and vice versa. This fact is apparently not mentioned (or at least not thoroughly discussed), and without some additional discussion and figures of actual data, the modeling results are very difficult to interpret. The canopy layer makes up <10% of the full boundary layer, so are findings of production rates, etc. normalized or corrected for this small contribution to the mean boundary layer concentrations? Presumably the observations used to constrain the model are vertically resolved, but no vertical profiles of the observations are shown to evaluate how representative the two boxes used for modeling actually are. Oxidation products formed below the canopy, or above, would cycle many times between the two regions over the course of a day, scrambling the signatures and likely diluting the effect of chemistry occurring within the canopy itself.

Author Response:

While our original aim was to consider specifically the changes (relative to above canopy) in gas- and particle-phase chemistry caused by canopy shading, we acknowledge that mixing within forest canopies substantially influences overall forest chemistry (Wolfe et al., 2011; Bryan et al., 2012).  Thus, we have added a parameterization of transport between the above- and below-canopy boxes as well as upward from the above-canopy box, as described subsequently.  This inclusion of transport also addresses the comment posted by B. Bohn (including citation of Bohn, 2006).

Measured and modeled in-canopy residence times vary substantially depending on the forest environment studied. For instance, Fuentes et al. (2007) report average residence times of ~8 minutes for a parcel emitted near the ground in a forest with a 26-m high canopy, while Farmer and Cohen (2008) calculate residence times of 1-7 minutes for a forest with a canopy height of only 5.7m. Maximum residence times of up to 50 minutes have been reported in tall forests (Strong et al., 2004). Transport back into the canopy is an even more complicated process, as coherent structures (i.e., sweeps of air downward), rather than simple turbulence, often produce the majority of scalar fluxes (Steiner et al., 2011).

Rather than simply selecting a reasonable characteristic residence time, we have run a sensitivity study to optimize it based on model output.  A cost function has been applied

to compare modeled and measured diurnal concentrations of methacrolein and methyl vinyl ketone (MACR+MVK) above and below the canopy, in order to determine ideal transport residence time values. In addition, we have included a diurnal vertical dilution rate based on the average of methanol and acetaldehyde above-canopy vertical loss calculated by the FORCAsT 1-D model for summertime conditions (Ashworth et al., 2016).

Resulting changes in manuscript:

The revised manuscript will show a subsection within the Methods that describes this transport parameterization in detail, and the revised Supplemental Information will display a figure to portray the cost function output.  In addition, a new sentence will emphasize that because the above-canopy model was based on conditions observed only a few meters above the canopy, the modeled concentrations of oxidation products and SOA cannot be assumed to exist throughout the entire mixed boundary layer.  This is appropriate for the emphasis of the paper on how daytime in-forest $NO_3$ chemistry impacts SOA.

2. Referee:

While the MCM is becoming more useful, it is still rather a dangerous model to use for SOA studies because the mechanisms are not complete at the 5% level of carbon mass, but that 5% might be 90% of carbon capable of forming SOA. For example, the MCM does not include auto-oxidation of monoterpenes (Ehn et al Nature 2014 and others) - thereby missing a likely significant fraction of SOA mass, as well as highly oxygenated nitrates (Lee et al PNAS 2016). How might this affect the conclusions?

Author Response:

Despite the detail included in and the recent advances that have been made to the MCM, we agree that only modeling the partitioning of MCM oxidation products may result in an under prediction of SOA (depsite the MCM being among the most complete mechanisms available). Rather than simply explaining the likely under-prediction, we have altered the model to include two mechanisms of aerosol formation other than simple partitioning of the MCM oxidation products: production of extremely low-volatility organic compounds (ELVOCs) formed from the autooxidation of α-pinene and isoprene (Ehn et al., 2014; Jokinen et al., 2015; Mentel et al., 2015) and reactive uptake of isoprene epoxydiol, glyoxal, and methylglyoxal onto aerosol surfaces (Paulot et al., 2009; Ervens and Volkamer, 2010; Lin et al., 2012; Li et al., 2015). ELVOCs are thought to constitute a substantial fraction of total SOA in environments with low overall OA mass loadings (Ehn et al., 2014), and the reactive uptake of isoprene products has been shown to constitute over half of total SOA in isoprene-dominated forests (Li et al., 2015).  In this way, we address both the under-prediction of total SOA addressed here and the high fraction of organic nitrate products contained within the simulated SOA in our original manuscript (see referee #1, point 5 below).

In terms of ELVOC production, α-pinene and isoprene are each assumed to produce one ELVOC product from oxidation by both $O_3$ and OH, following published yields for each of the four relevant reactions (Jokinen et al., 2015). The ELVOC products observed in chamber studies are often either $C_{10}$ monomers or $C_{19-20}$ dimers from α-pinene oxidation and $C_5$ monomers from isoprene oxidation (Jokinen et al., 2015); however, as monomers are generally observed with a mass spectral signal an order of magnitude higher than those of dimers, and in order to ensure that uncertainty in the modeling parameters results in under-prediction rather than over-prediction of ELVOC mass concentrations, both products are assumed to be monomers. These ELVOCs are highly oxidized species with multiple hydroperoxide moieties (Mentel et al., 2015) and have chemical formulas of $C_{10}H_{16}O_9$ and $C_5H_8O_8$ from α-pinene and isoprene oxidation, respectively. These specific products were selected based on their intensity in observed ELVOC mass spectra and the fact that their O:C ratios are generally representative of the average ELVOC product distributions observed (Ehn et al., 2014; Jokinen et al., 2015).

In order to model the reactive of uptake of isoprene epoxydiols, glyoxal, and methylglyoxal, we adopted the method of Li et al. (2015). The formula for the rate of uptake is dependent on the mass concentration, the thermal velocity of the species, the ambient aerosol surface area, and a reactive uptake coefficient (Li et al., 2015). Ambient aerosol surface area data during CABINEX were obtained from VanReken et al. (2015). In accordance with Li et al. (2015), glyoxal and methylgloxal were both assigned a dimensionless reactive uptake coefficient of 2.9 x $10^{-3}$, following the findings of Lin et al. (2012), while the reactive uptake of epoxydiols was assumed to be 0.5 x $10^{-3}$, representative of conditions modeled in Michigan.

Resulting changes in manuscript:

The revised manuscript will include a section in the Methods describing these changes in detail, and the results will be updated accordingly.

3. Referee:

Lack of deposition. Deposition of oxidation products appears not to be included. This seems rather problematic for interpreting the SOA formation potential. Deposition in the canopy will be much more significant than above the canopy, at least during nighttime, and during daytime deposition from both would significantly impact the available vapors for SOA formation, especially given the equilibrium partitioning assumption used for modeling SOA. With any horizontal wind through the canopy, there will presumably be a significant depositional sink given the proximity to canopy elements. What is the lifetime of a product formed in the canopy against deposition compared to mixing out of the canopy layer or to SOA formation? How was the vertical distribution of the condensation sink constrained?

Author Response:

Please see the earlier author response about ignorance of transport. Dry deposition of both gases and aerosol is now included in the modeling framework.

In the above-canopy model, deposition is assumed to occur onto the top of the canopy, and the resistance to deposition is determined using Meyers and Baldocchi (1988). Deposition velocities for each chemical species are calculated based on resistances from the quasi-laminar boundary layer and the leaf mesophyll, cuticular surfaces, and stomata. A thorough description of the particular equations used in this method can be found in Bryan et al. (2012). The above-canopy box height was assumed to vary diurnally based on the boundary layer values utilized in Giacopelli et al. (2005) for a previous box model of the PROPHET location. As our model does not calculate aerosol sizes, size distribution data obtained from VanReken et al. (2015) were used to calculate a volume-weighted average settling velocity of aerosol particles.

In the below-canopy model, deposition was assumed to occur to the ground and was modeled following the method of Gao et al. (1993). The box height was set to 6m (the assumed bottom of the canopy layer or top of the trunk space) for the entire diurnal period. In a sub-canopy environment, turbulence may result in upward transport and deposition onto foliage in the bottom of the canopy layer; however, as modeled concentrations of certain oxidation products (MACR+MVK and HCHO) are lower than those measured assuming deposition only occurs to the ground, no loss to foliage in the trunk space was considered.

Resulting changes in manuscript:

The revised manuscript will include a section within the Methods that describes the parameterization of deposition, and the results will be updated accordingly. The revised supplemental information will include a table that contains a description of the specific parameters used in the calculation of these resistances. For the majority of VOC oxidation products, these parameters, Henry's Law constants and diffusion coefficients, for example, were obtained from Nguyen et al. (2015). Data obtained elsewhere in the literature are specified as such in the supplemental information.

4. Referee:

Given 2 and 3 above, the discussion on SOA seems rather limited in its usefulness to actual SOA formation potential. In addition, more interesting would be to incorporate the lapse rate and vertical mixing impacts on SOA formation given the cycling of oxidation products across a 10K temperature gradient.

Response:

Having addressed issues 2 and 3, we believe model results with respect to SOA formation potential are now valid. In addition, we wish to stress that the aim is to evaluate impacts of shading on chemistry.

Resulting changes in manuscript:

None.

5. Referee:

If I understand correctly, RONO2 form ~45% of the SOA mass in the model. In the SE U.S., it was more like 3-10% (Xu et al PNAS 2015, Lee at al PNAS 2016). The authors cite Fry et al, and note consistency with that study, but, are the distributions of BVOC emissions at all similar between Manitou and UMBS? Presumably the high fraction of particulate RONO2 at Manitou was the dominance of monoterpenes, is that the case at UMBS?

Author Response:

We thank the reviewer for highlighting this discrepancy. We believe that the high fraction of $RONO_2$ within the SOA was largely the result of a both a lack of SOA production (by mechanisms not captured by simple reversible partitioning of MCM oxidation products) and a lack of nitrate hydrolysis within the SOA. The first of these issues was addressed above (reviewer #1, point #2). However, in order to improve the accuracy of the model, we have also added a mechanism for nitrate hydrolysis within the aerosol. The heterogeneous hydrolysis of organic nitrates has recently received significant attention. Multiple chamber experiments on the hydrolysis of both α-pinene and isoprene nitrates have been performed, and while many questions remain regarding the specific mechanisms of hydrolysis, these studies have defined lifetimes or loss rates of nitrates within organic aerosol, necessary parameters for modeling simple first order loss (Cole-Filipiak et al., 2010; Darer et al., 2011; Hu et al., 2011; Bean and Hildebrandt Ruiz, 2016; Rindelaub et al., 2015).

In order to characterize more accurately the nitrate content of the SOA, both isoprene and α-pinene nitrate products are assumed to undergo first-order loss within the aerosol. Isoprene nitrate loss is parameterized using the average lifetimes found by Hu et al. (2011). As primary and secondary nitrates are found to only slowly hydrolyze even under the most acidic conditions observed in aerosol ($\tau = 500$ hr at pH = 0), which are not likely to occur at PROPHET, those products have an effective hydrolysis loss rate of zero within the model (Hu et al., 2011). However, tertiary nitrates, which are efficiently hydrolyzed even at neutral pH, have an effective lifetime of 0.67 hours, corresponding to a first-order loss rate of $1.73 \times 10^{-5}$ s$^{-1}$ (Hu et al., 2011).

Two sets chamber experiments have been performed regarding hydrolysis of α-pinene organic nitrates within organic aerosol (Rindelaub et al., 2015; Bean and Hildebrandt Ruiz, 2016). Of these, only the study by Bean and Hildebrandt Ruiz (2016) quantified rates of nitrate hydrolysis within the aerosol. The hydrolysis rate was found to be highly dependent on RH; experiments with RH 20-60% produced a hydrolysis rate of 2 day$^{-1}$, while those with RH above 70% had rates as high as 7 day$^{-1}$ (Bean and Hildebrandt Ruiz,

2016). As RH values measured during CABINEX are generally above 60%, we utilized the rate of 7 day$^{-1}$, corresponding to an organic nitrate lifetime of 3.4 hr.

Resulting changes in manuscript:

The updated manuscript will include a short section within the Methods describing this change and will update the results accordingly.

6. Referee:

Odd model set up choices. The "urban" case, which apparently mixes data from Detroit and Houston is a bit odd. Why not make it purely hypothetical? I don't see why the authors fix the NO/NO2 ratio - that would seem to be a good metric to test the chemistry in the model as it is sensitive to total RO2, not just HO2 and O3. Moreover, as NOx increases the NO/NO2 ratio isn't going to be fixed, there are important feedbacks between O3, HO2, RO2, and NO concentrations that are important and why one would want to use a model in the first place. Fixing the NO/NO2 ratio makes nitrate formation rates a linear extrapolation with increasing NOx when the non-linear couplings of HOx and NOx might lead to a different result than found here.

Response:

We agree that the overall method of testing the sensitivity of the results could be better designed. In order to more reasonably test the sensitivity of forest environments to different concentrations of anthropogenic pollutants, we have removed the second and third model cases and replaced them with a more structured sensitivity analysis. This method independently modifies the $O_3$ and $NO_x$ concentrations by scaling the diurnal profiles of $O_3$, NO, and $NO_2$ under different NO/$NO_2$ ratios. As a result, we are able to evaluate the results under high pollutant concentrations when $NO_3$ formation is favored (high $NO_x$, high $NO_2$/NO ratio) and when $NO_3$ formation is suppressed (low $NO_x$, low $NO_2$/NO ratio).

Resulting changes in manuscript:

The revised manuscript will include an updated section on input data in the Methods and will include updated results accordingly.

REVIEWER #2

1. Referee:

The authors dedicate lengthy discussions to obvious and/or unsurprising results. Some examples include: (1) that isoprene oxidation is predominantly by reaction with OH is mentioned in the abstract and has a dedicated paragraph (p. 11, lines 23-29) and figure (Figure 6); (2) an entire section is dedicated to detailing the enhancement in daytime nitrate concentrations (Section 3.2, p. 10), but this can be reduced to 1-2 sentences pointing to the importance of photolysis; and (3) page 8, lines 30-34: that NO3 and O3 perturbations change NO3 is obvious and indicated in Equation (4).

Author Response:

We agree that shortening or removing these specific cases (which were included for completeness and proof of model performance) is an appropriate change. With regard to the section describing $NO_3$ concentrations, because $NO_3$ is sensitive to both photolysis rates and $O_3$ and $NO_x$ profiles, which are substantially different above and below the canopy, we considered it appropriate to include a short analysis highlighting the specific times when reductions in photolysis caused the elevation in $NO_3$ concentrations, rather than simply claim that this was the case during the middle of the day.

Resulting changes in manuscript:

The revised manuscript will show that:
- The mention of isoprene oxidation has been removed from the abstract, as has most of the original paragraph describing isoprene oxidation. Only one sentence has been included at the end of the previous paragraph.
- The section describing the daytime enhancement in $NO_3$ concentrations below the canopy has been reduced and added to the section regarding $NO_3$ concentrations.
- The section describing the $NO_3$ steady state analysis has been removed.

2. Referee:

Steps in the methodology are not well justified and so appear arbitrary. For example, the authors include observations from a site in Detroit and also appear to impose an imaginary forest canopy in the 0D simulations. The analysis of the CABINEX site under polluted conditions seems sufficient to show the effect of pollution on RONO2 and SOA formation above and below a forest canopy.

Author Response:

Please see response to Reviewer #1, Point #6.

3. Referee:

It is not clear from the abstract or concluding statements what the impact is of the results from this study. Are there any implications in this study for greening urban areas or for rapid urbanization (that is, rapid land cover change in and around urban areas)? In the Short Summary the authors mention that it is important to understand the impact of forest canopies on the oxidative fate and SOA formation of reactive VOCs "as forested areas downwind of urban areas (and therefore the residents) will be impacted by this phenomenon." How will they be impacted?

Author Response:

We acknowledge that a discussion of impacts of the study could be expanded. Ultimately, we conclude that the elevation of $NO_3$ below the canopy influences organic nitrate production rates substantially during the day, leading to a different suite of organic nitrate products above and below the canopy. The effect of elevated $NO_3$ concentrations on SOA formation becomes more significant under polluted conditions with a favorable $NO_2/NO$ ratio, which could indicate that forested areas on the outskirts of urban environments may experience substantially higher SOA mass loadings below the canopy than above the canopy, potentially affecting the health of local communities.

Resulting changes in manuscript:

Both the abstract and the conclusions in the updated manuscript will make the implications of the results more explicit.

SPECIFIC COMMENTS, REVIEWER #2

4. Referee:

Page 2, line 2: The start of the sentence "The most significant first-generation RONO2 formation mechanism" is ambiguous. Do the authors mean that most RONO2 is first-generation RONO2 or is this referring to a specific first-generation RONO2 species that dominates?

Author Response/Changes in Manuscript:

Because of the large number of changes incorporated into the model based on both reviewers' suggestions, especially with regard to the sensitivity analysis, this sentence will no longer be relevant.

5. Referee:

Page 2, lines 28-29: Will the 0D model in this study then also underestimate OA mass loadings?

Author Response/Changes in Manuscript:

Our model may underestimate OA mass loadings, as understanding of oxidation, partitioning, and particle-phase chemical mechanisms (oligomerization, the effect of liquid water) are not yet complete enough to ensure entirely accurate results. The aim here is to highlight the potential importance of $NO_3$ chemistry. However, the inclusion of recently discovered pathways for SOA formation (ELVOC formation and reactive uptake of isoprene products) reduces the amount of under-prediction. Furthermore, by ensuring that we under-predict rather than over-predict SOA mass loadings, we can be confident that our conclusions are conservative with respect to SOA formation. These points will be emphasized in the description of limitations of the SOA modeling in the updated manuscript.

6. Referee:

Page 3, lines 20-21: Are the authors referring to a previous study when they state: "It has been further hypothesized that"? If so, then please provide the reference.

Author Response/Changes in Manuscript:

The mentioned sentence has been altered to read, "Furthermore, the majority of these studies have modeled relatively remote locations, encouraging an evaluation of the sensitivity of modeled results to background pollutant concentrations."

7. Referee:

Page 3, line 37: Please briefly indicate for the reader what the differences are in BVOC (isoprene and alpha-pinene) reactivity and SOA formation.

Author Response/Changes in Manuscript:

The following sentences have been added to the manuscript at the appropriate location:

"For instance, isoprene is known to react predominately with OH, while α-pinene reacts substantially with all three major oxidants (Fuentes et al., 2007). The SOA yields measured during the photooxidation of isoprene are generally lower than those from α-pinene (0-0.053 for isoprene; 0.06-0.21 for α-pinene); however, their SOA yields are similar under conditions of $NO_3$ oxidation (0.04-0.238 for isoprene and 0.04-0.16 for α-pinene) (Spittler et al., 2006; Ng et al., 2007; Carlton et al., 2009). As α-pinene is much more likely to react with $NO_3$ than isoprene, this study highlights potential differences that increased concentrations of $NO_3$ can have on SOA formation from different VOCs."

8. Referee:

Page 4, lines 21-24: Please indicate what the effect is of not accounting for liquid water content and particle-phase reactions on SOA predicted by the model.

Author Response/Changes in Manuscript:

The following sentences have been added to the updated manuscript.

"As these particle-phase reactions can rapidly produce high MW, low volatility compounds, often more aerosol mass is produced in the environment than equilibrium partitioning of the gas-phase species would predict (Johnson et al., 2005; Kroll and Seinfeld, 2008). Furthermore, as increases in RH, and subsequent increases in the water content of SOA, are known to enhance the partitioning of organic species to the aerosol phase, the omission of this process also potentially leads to underestimation of the total amount of SOA formed (Hennigan et al., 2008). However, as aerosol liquid water is generally driven by inorganic aerosol components (Carlton and Turpin, 2013), which only comprised a small fraction of total aerosol mass during CABINEX, the overall effect of RH is predicted to be small for the conditions of this study (Malm and Day, 2001; Hennigan et al., 2008; VanReken et al., 2015). Therefore, the primary uncertainty in aerosol formation is related to the production of high MW compounds through particle-phase reactions, but these compounds are partially accounted for through the parameterization of ELVOC formation."

9. Referee:

Page 6, line 1: "...varies over the range of 0 to 0.17." This range includes night time, but more useful for the reader is the variability in this ratio during daylight hours when photolysis is occurring.

Author Response/Changes in Manuscript:

The above has been changed to "…varies over the range of 0.05 to 0.17 during daylight hours."

10. Referee:

Page 6, lines 1-2: "Model input data are further described in the SI". There is no description of the model input data in the SI; only figures and captions.

Author Response/Changes in Manuscript:

The above has been changed to, "Model input data are shown in the figures in the SI."

11. Referee:

Page 6, line 13: "In the second modeled case...". What was the first modeled case? This is not systematically presented in the methods section.

Author Response/Changes in Manuscript:

As our method of testing the sensitivity of the results has changed from investigating specific scenarios to evaluating the results more generally using a sensitivity analysis, this sentence no longer applies in the updated manuscript.

12. Referee:

Page 6, lines 34-35: Please indicate in the text why Houston sites were selected to obtain NO2/NO ratios for Detroit. Are NO2/NO ratios reasonably similar for all urban sites?

Author Response/Changes in Manuscript:

As with the last comment, this comment no longer applies in the updated setup of the study.

13. Referee:

Page 7, line 4: "The Detroit data display...". Please indicate in the text where this data is displayed.

Author Response/Changes in Manuscript:

As with the last comment, this comment no longer applies in the updated setup of the study.

14. Referee:

Page 7, line 12: Please point the reader to Figure S4 at the start of paragraph, rather than midway through. This provides context for the discussion.

Author Response/Changes in Manuscript:

The reference to the Figure S4 has been added to the end of the first sentence in the paragraph.

15. Referee:

Page 7, lines 14-15: "The model tends to under predict nighttime OH concentrations". Is this after taking into consideration measurement interference?

Author Response/Changes in Manuscript:

The interference involved in the IU-FAGE measurement affected only measured $HO_2$ concentrations during CABINEX (Griffith et al., 2013). This point is clarified within the section in the updated manuscript.

16. Referee:

Page 8, lines 4-5: It's not apparent how the results support an isoprene-derived measurement interference. The slope is near unity (Figure S5(b)) when the model does not include interfering isoprene RO2, but less than unity (slope = 0.7) when it does. This would suggest that the interference is negligible.

Author Response/Changes in Manuscript:

In this case, because the isoprene-derived interference has been well-characterized elsewhere (Fuchs et al., 2011; Griffith et al., 2013), we assume that the agreement seen between our model and the measurements in terms of $HO_2$ concentrations is due to an over-prediction of $HO_2$ by our model. The fact that the diurnal profiles are so similar (when not including isoprene $RO_2$) is therefore likely somewhat of a coincidence. We will clarify this fact within the updated manuscript; however, the agreement between our model and measured concentrations of $HO_x$ is still well within the range of previously published models (Lelieveld et al., 2008; Pugh et al., 2010; Stavrakou et al., 2010).

17. Referee:

Page 12, lines 29-32: More appropriate to compare your total OA (and not SOA only) with total OA from Delia (2004), as the contribution of SOA to total OA is known (page 12, line 37 and page 13, line 1).

Author Response/Changes in Manuscript:

We have altered the sentence to compare our SOA+POA with the OA measurements made by Delia (2004).

18. Referee:

Page 13, lines 8-9: Does the difference in photolysis above and below the canopy impact SOA formation? It is not apparent that this has been tested in the sensitivity simulations (Figure S6).

Author Response/Changes in Manuscript:

It is clear that the photolysis of $NO_3$ impacts SOA above and below the canopy. However, it also is likely that photolysis of oxidation products themselves affects SOA formation. Based on the assumption that the above-canopy radiation is the upper bound, we have included an investigation of how reducing photolysis rates affects SOA in the sensitivity analysis section of the SOA results.

19. Referee:

The authors provide labels for above canopy and below canopy data in panel "(a)" of Figures 2 and S2 that seems to suggest the dark lines are for below canopy data and the

lighter lines for above canopy data. If this is the case, the authors should clarify this convention in the figure captions.

Author Response/Changes in Manuscript:

The reviewer's interpretation of the lines is correct. We have edited the figure captions to clarify the coloring conventions.

References Cited

Ashworth, K., Chung, S. H., McKinnet, K. A., Liu, Y., Munger, B. J., Martin, S. T., and Steiner, A. L.: Modelling bi-directional fluxes of methanol and acetaldehyde with the FORCAsT canopy exchange model, Atmos. Chem. Phys. Discuss., doi:10.5194/acp-2016-522, in review, 2016.

Bean, J. and Hildebrandt Ruiz, L.: Gas-particle partitioning and hydrolysis of organic nitrates formed from the oxidation of α-pinene in environmental chamber experiments, Atmos. Chem. Phys., 16, 2175-2184, doi:10.5194/acp-16-2175-2016, 2016.

Bohn, B.: Solar spectral actinic flux and photolysis frequency measurements in a deciduous forest, J. Geophys. Res., 111, D15303, doi:10.1029/2005JD006902, 2006.

Bryan, A. M., Bertman, S. B., Carroll, M. A., Dusanter, S., Edwards, G. D., Forkel, R., Griffith, S., Guenther, A. B., Hansen, R. F., Helmig, D., Jobson, B. T., Keutsch, F. N., Lefer, B. L., Pressley, S. N., Shepson, P. B., Stevens, P. S., and Steiner, A. L.: In canopy gas-phase chemistry during CABINEX 2009: sensitivity of a 1-D canopy model to vertical mixing and isoprene chemistry, Atmos. Chem. Phys., 12, 8829–8849, doi:10.5194/acp-12-8829- 2012, 2012.

Carlton, A. G., and Turpin, B.J.: Particle partitioning potential of organic compounds is highest in the Eastern US and driven by anthropogenic water, Atmos. Chem. Phys., 13, 10203-10214, 2013.

Carlton, A. G., Wiedinmyer, C., and Kroll, J. H.: A review of Secondary Organic Aerosol (SOA) formation from isoprene, Atmos. Chem. Phys., 9, 4987-5005, 2009.

Cole-Filipiak, N. C., O'Connor, A. E., and Elrod, M. J.: Kinetics of the hydrolysis of atmospherically relevant isoprene-derived hydroxy epoxides, Environ. Sci. Technol., 44, 6718–6723, 2010.

Darer, A. I., Cole-Filipiak, N. C., O'Connor, A. E., and Elrod, M. J.: Formation and stability of atmospherically relevant isoprene derived organosulfates and organonitrates., Environ. Sci. Technol., 45, 1895–902, doi:10.1021/es103797z, 2011.

Delia, A.: Real-Time Measurements of Non-Refractory Particle Composition and Interactions at Rural and Semi-Rural Sites, PhD Thesis, University of Colorado-Boulder, 2004.

Ehn, M., Thornton, J. A., Kleist, E., Sipila, M., Junninen, H., Pullinen, I., Springer, M., Rubach, F., Tillmann, R., Lee, B., LopezHilfiker, F., Andres, S., Acir, I. H., Rissanen, M., Jokinen, T., Schobesberger, S., Kangasluoma, J., Kontkanen, J., Nieminen, T., Kurten, T., Nielsen, L. B., Jorgensen, S., Kjaergaard, H. G., Canagaratna, M., Dal Maso, M., Berndt, T., Petaja, T., Wahner, A., Kerminen, V. M., Kulmala, M., Worsnop, D. R., Wildt, J., and Mentel, T. F.: A large source of low-volatility secondary organic aerosol, Nature, 506, 476–479, 2014.

Ervens, B., and Volkamer, R.: Glyoxal processing by aerosol multiphase chemistry: towards a kinetic modeling framework of secondary organic aerosol formation in aqueous particles. Atmos. Chem. Phys. 10, 8219e8244, 2010.

Farmer, D. K., and Cohen, R. C.: Observations of $HNO_3$, $\Sigma AN$, $\Sigma PN$, and $NO_2$ fluxes: evidence for rapid $HO_x$ chemistry within a pine forest canopy, Atmos. Chem. Phys., 8, 3899-3917, 2008.

Fuchs, H., Bohn, B., Hofzumahaus, A., Holland, F., Lu, K. D., Nehr, S., Rohrer, F. and Wahner, A.: Detection of HO2 by laser-induced fluorescence: calibration and interferences from RO2 radicals, Atmos. Meas. Tech., 4, 1209-1225, doi:10.5194/amt-4-1209-2011, 2011.

Fuentes, J. D., Wang, D., Bowling, D. R., Potosnak, M., Monson, R.K., Goliff, W. S., and Stockwell, W. R.: Biogenic Hydrocarbon Chemistry within and Above a Mixed Deciduous Forest, J. Atmos. Chem., 56, 165-185, doi:10.1007/s10874-006-9048-4, 2007.

Gao, W., Wesely, M., and Doskey, P.: Numerical modeling of the turbulent diffusion and chemistry of $NO_x$, $O_3$, isoprene, and other reactive trace gases in and above a forest canopy, J. Geophys. Res., 98, D10, 18,339-18,353, 1993.

Giacopelli, P., Ford, K., Espada, C., and Shepson, P. B.: Comparison of the measured and simulated isoprene nitrate distributions above a forest canopy., J. Geophys. Res., 110, D01304, doi:10.1029/2004JD005123, 2005.

Griffith, S. M., Hansen, R. F., Dusanter, S., Stevens, P. S., Alaghmand, M., Bertman, S. B., Carroll, M. A., Erickson, M., Galloway, M., Grossberg, N., Hottle, J., Hou, J., Jobson, B. T., Kammrath, A., Keutsch, F. N., Lefer, B. L., Mielke, L. H., O'Brien, A., Shepson, P. B., Thurlow, M., Wallace, W., Zhang, N., and Zhou, X. L.: OH and HO2 radical chemistry during PROPHET 2008 and CABINEX 2009 – Part 1: Measurements and model comparison, Atmos. Chem. Phys., 13, 5403-5423, doi:10.5194/acp-13-5403-2013, 2013.

Hennigan, C. J., Bergin, M. H., Dibb, J. E., and Weber, R. J.: Enhanced secondary organic aerosol formation due to water uptake by fine particles, Geophys. Res. Lett. 35, L18801, 2008.

Hu, K., Darer, A., and Elrod, M.: Thermodynamics and kinetics of the hydrolysis of atmospherically relevant organonitrates and organosulfates, Atmos. Chem. Phys., 11, 8307-8320, doi:10.5194/acp-11-8307-2011, 2011.

Johnson, D., Jenkin, M.E., Wirtz, K., and Martin-Reviejo, M.: Simulating the formation of secondary organic aerosol from the photooxidation of aromatic hydrocarbons. Environmental Chemistry 2, 35–48, 2005.

Jokinen, T., Berndt, T., Makkonen, R., Kerminen, V.-M., Junninen, H., Paasonen, P., Startmann, F., Herrmann, H., Guenther, A., Worsnop, D. R., Kulmala, M., Ehn, M., and Sipilä, M.: Production of extremely low-volatile organic compounds from biogenic emissions: measured yields and atmospheric implications, P. Natl. Acad. Sci. USA, 112, 7123–7128, 2015.

Kroll, J.H. and J.H. Seinfeld: Chemistry of secondary organic aerosol: Formation and evolution of low volatility organics in the atmosphere, Atmos. Environ. 42, 3593–3624, doi:10.1016/j.atmosenv.2008.01.003, 2008.

Lelieveld, J., Butler, T. M., Crowley, J. N., Dillon, T. J., Fischer, H., Ganzeveld, L., Harder, H., Lawrence, M. G., Martinez, M., Taraborrelli, D., and Williams, J.: Atmospheric oxidation capacity sustained by a tropical forest, Nature, 452, 737–740, doi:10.1038/nature06870, 2008.

Li, J., Cleveland, M., Ziemba, L. D., Griffin, R. J., Barsanti, K. C., Pankow, J. F., and Ying, Q.: Modeling regional secondary organic aerosol using the Master Chemical Mechanism, Atmos. Environ., 102, 52-61, 2015.

Lin, Y.H., Zhang, Z.F., Docherty, K.S., Zhang, H.F., Budisulistiorini, S.H., Rubitschun, C.L., Shaw, S.L., Knipping, E.M., Edgerton, E.S., Kleindienst, T.E., Gold, A., and Surratt, J.D.: Isoprene epoxydiols as precursors to secondary organic aerosol formation: acid-catalyzed reactive uptake studies with authentic compounds. Environ. Sci. Technol. 46, 250e258, 2012.

Malm, W. C., and Day, D. E.: Estimates of aerosol species scattering characteristics as a function of relative humidity, Atmos. Environ., 35(16), 2845 – 2860, 2001.

Mentel, T. F., Springer, M., Ehn, M., Kleist, E., Pullinen, I., Kurtén, T., Rissanen, M., Wahner, A., and Wildt, J.: Formation of highly oxidized multifunctional compounds: autoxidation of peroxy radicals formed in the ozonolysis of alkenes – deduced from structure–product relationships, Atmos. Chem. Phys., 15, 6745–6765, doi:10.5194/acp-15-6745-2015, 2015.

Meyers, T. P. and Baldocchi, D. D.: A comparison of models for deriving dry deposition fluxes of O3 and SO2 to a forest canopy, Tellus B, 40B, 270–284, doi:10.1111/j.1600- 0889.1988.tb00297.x, 1988.

Ng, N. L., Chhabra, P. S., Chan, A. W. H., Surratt, J. D., Kroll, J. H., Kwan, A. J., McCabe, D. C., Wennberg, P. O., Sorooshian, A., Murphy, S. M., Dalleska, N. F., Flagan, R. C., and Seinfeld, J. H.: Effect of NOx level on secondary organic aerosol (SOA) formation from the photooxidation of terpenes, Atmos. Chem. Phys., 7, 5159–5174, doi:10.5194/acp-7-5159-2007, 2007;.

Nguyen, T., Crounse, J., Teng, A., St. Clair, J., Paulot, F., Wolfe, G., and Wennberg, P.: Rapid deposition of oxidized biogenic compounds to a temperate forest, P. Natl. Acad. Scie, 112, 5, 392-401, doi: 10.1073/pnas.1418702112, 2015.

[revised manuscript text omitted]

Rob Griffin 11/4/2016 9:05 AM
Deleted: If this result can be confirmed by more sophisticated 1-D models that can more accurately describe the suite of BVOCs within the forest, transport through the forest and associated deposition, and hydrolysis losses, regional modeling may need to consider transport in order to accurately predict SOA mass loadings of semi-volatile oxidation products.

Rob Griffin 11/17/2016 3:10 PM

Rob Griffin 11/17/2016 3:10 PM

Rob Griffin 11/17/2016 3:10 PM

Rob Griffin 11/17/2016 3:11 PM

Rob Griffin 11/17/2016 3:11 PM

Rob Griffin 11/17/2016 3:11 PM

Rob Griffin 11/17/2016 3:11 PM

Ben Schulze 11/7/2016 1:29 PM

**Copyright Statement:**

Authors grant Copernicus Publications a licence to publish the article and identify itself as the original publisher.

Authors grant Copernicus Publications commercial rights to produce hardcopy volumes of the journal for purchase by libraries and individuals.

Authors grant any third party the right to use the article freely under the stipulation that the original authors are given credit and the appropriate citation details are mentioned.

The article is distributed under the Creative Commons Attribution 3.0  License. Unless otherwise stated, associated published material is distributed under the same licence.

**Code Availability:**

The MATLAB code associated with this manuscript is available upon request.

**Data Availability:**

The compiled datasets used to produce each figure within this manuscript are available as Igor Pro files upon request.

**Author Contribution:**

B.C. Schulze developed the model, performed simulations and data analysis, and wrote the manuscript. H.W. Wallace and R.J. Griffin assisted heavily with model development, data analysis, and manuscript editing. H.W. Wallace, J. H. Flynn, B.L. Lefer, M.H. Erickson, B.T. Jobson, S. Dusanter, S.M. Griffith, R.F. Hansen, T. VanReken, and P.S. Stevens performed atmospheric measurements during CABINEX that were used as model inputs. J.H. Flynn, P.S. Stevens, S. Dusanter, S.M. Griffith, and R.F. Hansen provided helpful comments and edits.

**Acknowledgements:**

We would like to acknowledge the National Science Foundation for funding this work with grant numbers AGS-0904214 and AGS-0904167.

Ben Schulze 11/15/2016 3:01 PM

[revised manuscript text omitted]

---

## Author Response (AR2)

Response to Editor Comments

Comments from the Editor highlighted in green, with response directly following. A text version of the main body with changes tracked is included. A new version of the SI has been uploaded.

page numbers are those in the author's response PDF)
1. P19, abstract: should be (8:00 – 20:00 LT), to be explicit.

   1. Added (LT) to be explicit.

2. P19, abstract: perhaps expand "suburban environment" to "suburban and peri-urban forest environments"?

   2. Expanded to suburban and peri-urban forest environments

3. P19, abstract: "relative to above the canopy" – add "in the model".

   3. Added "in the model"

4. P22, line 3: Matlab is the computing environment used, but a few more details are needed: which implicit Matlab subroutines were used, or was a bespoke ODE solver written? Delete "with numerical differentiation formulas". If there is no inorganic chemistry in the model, you should state this here as well as in section 2.5.
Equations 1 and 3. Use the same symbol for the same quantity (G_i and M_{air}) and please disambiguate gamma_i (e.g., use a_i for activity).

   4. Added that the ode15s solver was used…

The model uses a variable-order ordinary differential equation solver **(MATLAB® ODE solver ode15s)** to solve the underlying system of differential equations. One day of model spin-up was used for each analysis (Pratt et al., 2012).

Cont: Added that standard inorganic reactions from the MCM were used

The oxidation mechanisms of α-pinene and isoprene were obtained directly from the Master Chemical Mechanism (MCM v3.2, via website: http://mcm.leeds.ac.uk/MCM, Jenkin et al., 1997; Saunders et al., 2003**), as well as the standard gas-phase inorganic reactions included in the MCM**, resulting in a system of 2260 chemical reactions with 712 chemical species.

Cont: Changed the symbols so that gas-phase mass concentration is $G_i$ in both equations, $\alpha_i$ is the activity coefficient, and $\gamma_i$ is the reactive uptake coefficient.

5. P28, line 1. Please clarify "diel median values". To me, this means that each inorganic parameter has a single value assigned to it, and these parameter values change once a day, but it could be taken to mean that you use an average diurnal cycle fitted to each parameter, and that seems to be the case given Fig S3. In either case, it should be stated when the parameter values are updated.

5. Text now reads: The model was constrained by diel median values of hydrogen oxides ($HO_x = OH$ + hydroperoxy radical ($HO_2$)), NO, $NO_2$ $O_3$, $\alpha$-pinene (monoterpenes), isoprene, formaldehyde (HCHO), and the photolysis rate of $NO_2$. Each of these species was therefore assigned a median measured value at every hour, and values used for each model time step (e.g. 8:03) were determined by linear interpolation between hourly values.

6. Figure S3: measured median values should also show percentiles (25th and 75th or 10 and 90th, whichever feels more appropriate).

6. Figure has been updated.

7. P28: "throughout the diel period" – does this mean throughout the day, or thoughout the period used to derive the diurnal average?

7. Changed to "throughout the day".

8. P37, para 2: please add 'in the model' to 'majority of total SOA'.

   8. Added "in the model".

[revised manuscript text omitted]

Authors grant Copernicus Publications a licence to publish the article and identify itself as the original publisher.

Authors grant Copernicus Publications commercial rights to produce hardcopy volumes of the journal for purchase by libraries and individuals.

Authors grant any third party the right to use the article freely under the stipulation that the original authors are given credit and the appropriate citation details are mentioned.

The article is distributed under the Creative Commons Attribution 3.0  License. Unless otherwise stated, associated published material is distributed under the same licence.

[revised manuscript text omitted]